# Dual function of the PI3K-Akt-mTORC1 axis in myelination of the peripheral nervous system

**Gianluca Figlia, Camilla Norrmén, Jorge A Pereira, Daniel Gerber, Ueli Suter***

Department of Biology, Institute of Molecular Health Sciences, Swiss Federal Institute of Technology, Zürich, Switzerland

**Abstract** Myelination is a biosynthetically demanding process in which mTORC1, the gatekeeper of anabolism, occupies a privileged regulatory position. We have shown previously that loss of mTORC1 function in Schwann cells (SCs) hampers myelination. Here, we genetically disrupted key inhibitory components upstream of mTORC1, TSC1 or PTEN, in mouse SC development, adult homeostasis, and nerve injury. Surprisingly, the resulting mTORC1 hyperactivity led to markedly delayed onset of both developmental myelination and remyelination after injury. However, if mTORC1 was hyperactivated after myelination onset, radial hypermyelination was observed. At early developmental stages, physiologically high PI3K-Akt-mTORC1 signaling suppresses expression of Krox20 (Egr2), the master regulator of PNS myelination. This effect is mediated by S6K and contributes to control mechanisms that keep SCs in a not-fully differentiated state to ensure proper timing of myelination initiation. An ensuing decline in mTORC1 activity is crucial to allow myelination to start, while remaining mTORC1 activity drives myelin growth.
DOI: https://doi.org/10.7554/eLife.29241.001

## Introduction

Myelin is a lipid-rich membrane specialization of Schwann cells (SCs) and oligodendrocytes (OLs) winding around axons to accelerate the propagation of action potentials (*Hartline and Colman, 2007*). The geometrical properties of each myelin segment, such as its thickness and length, are essential to achieve fast and accurate conduction along myelinated fibers (*Waxman, 1980*). Conversely, myelin geometry is deranged in numerous diseases of the nervous system, including neuropathies and multiple sclerosis. To produce myelin segments with correct thickness and length, the plasma membrane of myelinating cells undergoes a remarkable, highly regulated expansion. It was estimated that the SC membrane has to expand several thousand fold during myelination (*Chrast et al., 2011*; *Webster, 1971*) and that OLs produce myelin equal to three times their own weight every day (*Norton and Poduslo, 1973*). Consequently, myelination is an anabolically demanding task.

The mechanistic target-of-rapamycin (mTOR), a serine-threonine protein kinase, is the core component of mTOR complex 1 (mTORC1) and mTOR complex 2 (mTORC2). Among them, mTORC1 is the gatekeeper of multiple anabolic reactions, including mRNA translation and biosynthesis of lipids, purines, and pyrimidines (*Ben-Sahra et al., 2013*; *Ben-Sahra et al., 2016*; *Porstmann et al., 2008*; *Thoreen et al., 2012*). The mTORC1 pathway is tightly connected to PI3K-Akt signaling downstream of growth factors. Upon growth factor stimulation, Akt activates mTORC1 by phosphorylating inhibitors of mTORC1, the TSC complex and PRAS40 (*Inoki et al., 2002*; *Manning et al., 2002*; *Sancak et al., 2007*; *Vander Haar et al., 2007*). The TSC complex is composed of the subunits TSC1, TSC2, and TBC1D7 (*Dibble et al., 2012*), and functions as a GTPase-activating unit towards the small GTPase Rheb, a potent activator of mTORC1 when loaded with GTP (*Tee et al., 2003*).

*For correspondence:
usuter@cell.biol.ethz.ch

Competing interests: The authors declare that no competing interests exist.

**eLife digest** Neurons transmit electrical impulses throughout our body along cable-like structures called axons. Similar to electric cables, the axons are enveloped in an insulating sheath called myelin, which makes the impulses travel faster down the axon. If myelin does not form correctly or gets lost later in life, it can lead to muscle weakness and numbness.

Myelin is produced in nerves by specialized cells called the Schwann cells, which wrap around the axons several times to create a thick myelin sheath. The signaling complex called mTORC1 plays an important role in this process. A study in 2014 showed that when mTORC1 was inactive, the myelin sheath was abnormally thin. However, it was not known whether increased mTORC1 would force Schwann cells to produce more myelin.

Now, Figlia et al. – including some of the researchers involved in the 2014 study – used genetically modified mice to manipulate two proteins known to control the activity of mTORC1. When both proteins were removed, either individually or in combination, mTORC1 activity was higher than normal. However, in developing nerve cells, high levels of mTORC1 did not cause the young Schwann cells to produce more myelin, but rather stopped them to become the specialized cells that wrap around axons. Figlia et al. then increased mTORC1 levels after Schwann cells had already started wrapping around axons. In this case, a high activity of mTORC1 resulted in thicker myelin. This suggests that a normal development of a nerve and ultimately the thickness of the myelin sheath depend on when and how much of mTORC1 is available.

This discovery could help to develop new therapies for myelin diseases. Increasing the activity of mTORC1 in Schwann cells after they have started wrapping may boost myelin production in diseases in which the myelin sheath is too thin. Conversely, inhibiting mTORC1 could help in situations when the Schwann cells cannot develop properly.

DOI: https://doi.org/10.7554/eLife.29241.002

Accordingly, disruption of the TSC complex causes mTORC1 hyperactivation in a wide range of tissues and cell types (*Byles et al., 2013*; *Castets et al., 2013*; *Kwiatkowski et al., 2002*).

Consistent with the intense anabolic challenge posed by myelination, the PI3K-Akt-mTORC1 axis has emerged as a fundamental player in PNS and CNS myelination (*Taveggia, 2016*; *Wood et al., 2013*). Genetic disruption of mTORC1 in SCs or OLs impaired myelination, demonstrating that mTORC1 function is required for this process (*Bercury et al., 2014*; *Lebrun-Julien et al., 2014*; *Norrmén et al., 2014*; *Sherman et al., 2012*; *Wahl et al., 2014*; *Zou et al., 2014*). However, studies aimed at examining the consequences of increased mTORC1 activity, predicted to augment myelin growth, have yielded conflicting results. Overexpression of constitutively active Akt or hyperactivation of the PI3K-Akt pathway by deletion of PTEN in OLs caused hypermyelination (*Flores et al., 2008*; *Goebbels et al., 2010*), while deleting TSC1 was detrimental to OL myelination (*Jiang et al., 2016*; *Lebrun-Julien et al., 2014*). In SCs, hyperactivation of the PI3K-Akt pathway with various approaches did not lead to univocal results (*Cotter et al., 2010*; *Domènech-Estévez et al., 2016*; *Flores et al., 2008*; *Goebbels et al., 2012*). Thus, we aimed here at elucidating the functional roles of mTORC1 activation in SCs during various steps of development, in homeostasis, and after injury. Based on our results and reconciling also previous reports, we propose a model suggesting that mTORC1 signaling exerts multiple distinct roles at different stages of SC differentiation.

## Results

### High mTORC1 signaling due to TSC1 deficiency in SCs delays the onset of myelination

Using a loss-of-function approach, we have previously discovered that mTORC1 – but not mTORC2 – promotes myelin growth in the PNS (*Norrmén et al., 2014*). To further define the role of mTORC1 in myelination, we have now pursued the converse approach, hyperactivating mTORC1 by conditional ablation of TSC1, a strategy that destabilizes the TSC complex (*Dibble et al., 2012*). SC-specific TSC1 mutants were generated by crossing mice harboring a floxed allele of *Tsc1* (*Kwiatkowski et al., 2002*) with mice expressing a *Cre* transgene under control of regulatory

sequences of the *Dhh* gene (*Jaegle et al., 2003*) (*Dhh^Cre^:Tsc1*^KO^). We first confirmed, by western blot analysis of postnatal day 5 (P5) sciatic nerves, that TSC1 was successfully depleted (*Figure 1a*, *Figure 1—figure supplement 1a*). We then assessed mTORC1 activity by analyzing the phosphorylation levels of its downstream effectors. As expected, phosphorylation of two well-established mTORC1 targets, S6K and 4EBP1 (*Hay and Sonenberg, 2004*), was increased in TSC1-mutant nerves, together with phospho-S6^S235/236^ levels, a target of S6K (*Figure 1a,b*, *Figure 1—figure supplement 1a*). As additional evidence of mTORC1 hyperactivation, we found that cultured mutant SCs isolated from either dorsal root ganglia (DRG) or postnatal nerves were enlarged, consistent with the prominent role of this pathway in cell size control (*Lloyd, 2013*) (*Figure 1c*, *Figure 1—figure supplement 2a,b*).

Next, we assessed the extent of myelination by electron microscopy (EM). Surprisingly, P5 *Dhh^Cre^:Tsc1*^KO^ nerves exhibited a strong reduction in myelinated fibers due to an arrest of most SCs at the promyelinating stage (*Figure 1d,e*). No overt defect in radial sorting was evident, thus indicating a *bona fide* impairment in the onset of myelination. The percentage of myelinated fibers progressively increased with time, almost doubling by P14. By P60, most fibers were ultimately myelinated, although occasional promyelinating SCs were still present (*Figure 1d,e*). Additionally, the myelinated nerve fibers were hypomyelinated, presumably as a consequence of delayed onset of myelination (*Figure 1d*; for quantification, see Figure 6l). Impaired SC differentiation was reflected in reduced levels of myelin protein P0, while cJun and Oct6 – both highly expressed in promyelinating SCs – were upregulated (*Figure 1—figure supplement 2c,d*). Consistent with a failure of mutant cells to promptly differentiate, we also detected an increase in proliferating Sox10-positive SCs and, consequently, we found overall more SCs (P3; *Figure 1f–h*).

Non-mTORC1 related functions of the TSC complex have been reported (*Neuman and Henske, 2011*). Thus, we assessed whether the phenotype of *Dhh^Cre^:Tsc1*^KO^ mice was genuinely due to high mTORC1 activity by treating mice with the mTORC1-inhibiting drug rapamycin. Rapamycin was administered at P3 and P4 and the mice analyzed at P5. Despite the short course of treatment, we found a robust increase in myelinated fibers in rapamycin-treated, compared to vehicle-only treated, mutant nerves (*Figure 1i,j*). The morphological rescue was paralleled by a decrease in cJun and an increase in P0, roughly back to the levels of control nerves, together with the expected suppression of S6K phosphorylation at the mTORC1-sensitive site T389 (*Figure 1k*, *Figure 1—figure supplement 1b*). In line with the in vivo results, we found that in vitro myelination of TSC1 mutant-derived DRG explant cultures was strongly defective, but could be remarkably improved by acute treatment with rapamycin, consistent with a PNS-autonomous origin of the in vivo rescue (*Figure 1—figure supplement 2e,f*).

Collectively, our data show that high mTORC1 activity following deletion of TSC1 in SCs has, paradoxically, a detrimental effect on PNS myelination by delaying the transition from promyelinating to myelinating SCs.

## Defective SC myelination as a result of TSC1 deletion is not due to feedback inhibition of the PI3K-Akt pathway

The PI3K-Akt pathway is a key upstream driver of mTORC1 and, in turn, mTORC1 activation dampens the PI3K-Akt pathway through multiple inhibitory feedback loops (*Efeyan and Sabatini, 2010*). To explain the discrepancy between the effects of high mTORC1 signaling in SCs lacking TSC1 and the positive role generally attributed to PI3K-Akt signaling in PNS myelination (*Taveggia, 2016*), we reasoned that overactive mTORC1 may suppress the PI3K-Akt pathway, and consequently SC differentiation, via the aforementioned feedback loops. To test this hypothesis, we assessed phosphorylation of Akt and the upstream receptor ErbB2 in *Dhh^Cre^:Tsc1*^KO^ nerves. Since the MAPK pathway can also be subjected to feedback inhibition by mTORC1 (*Carracedo et al., 2008*), and considering the crucial functions of this pathway in SC biology (*Ishii et al., 2013*; *Newbern et al., 2011*; *Sheean et al., 2014*), we also examined the phosphorylation status of Erk1/2. No major changes in ErbB2 or Erk1/2 phosphorylation could be detected (*Figure 2a*, *Figure 2—figure supplement 1a*). By contrast, we observed a strong reduction in Akt phosphorylation at both T308 and S473, together with a global decrease of Akt substrates phosphorylation (*Figure 2a*, *Figure 2—figure supplement 2a* and *Figure 2—figure supplement 1a*). Moreover, pharmacological inhibition of the PI3K-Akt axis, but not the Mek-Erk axis, abolished S6 phosphorylation upon neureugulin-1 stimulation of primary SCs, although some minor contribution of the Mek-Erk axis to mTORC1 activation

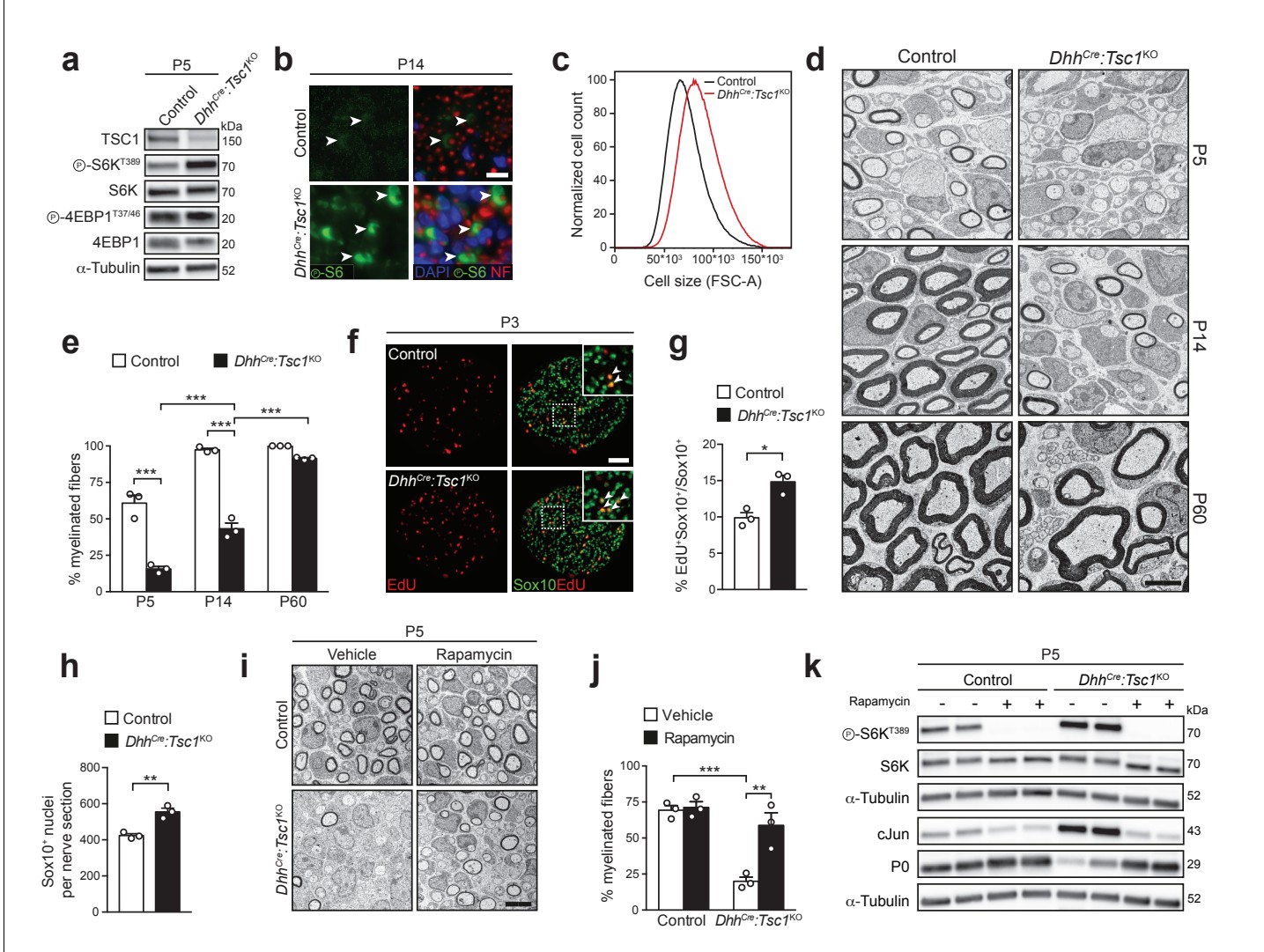

**Figure 1.** High mTORC1 signaling following TSC1 deletion delays SC myelination. (a) Western blot analysis, sciatic nerve lysates of P5 controls and littermates *Dhh^Cre^:Tsc1^KO^* (mutant) animals (n = 3 mice/genotype). Quantification: *Figure 1—figure supplement 1a*. (b) Immunostaining, P14 sciatic nerve cross sections (n = 3 control/4 mutant mice). Arrowheads indicate phospho-S6 positive cells. NF: neurofilament. Nuclei: DAPI. Scale bar: 10 µm, refers to whole panel. (c) Flow-cytometry profile of DRG-derived SCs (n = 4 individual experiments with four independent cell preparations). A representative experiment is shown. FSC-A: Forward Scatter Area. (d) Electron micrographs of sciatic nerve cross sections of controls and littermate mutants at different time points (n = 3 mice/genotype). Scale bar: 4 µm, refers to whole panel. (e) Myelinated fibers relative to all sorted fibers. Bar heights: Mean; error bars: s.e.m. (n = 3 mice/genotype, two-way ANOVA with Tukey's multiple comparisons test, $F_{(2,12)}$ = 37.09, p<0.0001; $p_{P5\ control\ vs\ KO}$ <0.0001, $p_{P14\ control\ vs\ KO}$ <0.0001, $p_{P5\ vs\ P14\ KO}$ = 0.0002, $p_{P14\ vs\ P60\ KO}$ < 0.0001). At least 150 sorted fibers per sample were counted (random EM fields). (f) EdU-labeling and Sox10 immunostaining on P3 sciatic nerve cross sections (n = 3 mice/genotype). Arrowheads indicate Sox10/EdU double-positive SC nuclei. Scale bar: 50 µm, refers to whole panel. (g,h) Quantifications of f. Sox10/EdU double-positive of Sox10-positive nuclei (g), Sox10-positive nuclei per sciatic nerve cross section (h). Bar heights: Mean; error bars: s.e.m. (n = 3 mice/genotype; two-tailed unpaired Student's t-test; g, $t_{(4)}$ = 4.407, p=0.0116; h, $t_{(4)}$ = 5.77, p=0.0045). (i) Electron micrographs of P5 sciatic nerves from control and mutants treated with vehicle or rapamycin. Scale bar: 4 µm, refers to whole panel. (j) Quantification of i. Myelinated fibers relative to all sorted fibers. At least 150 sorted fibers per sample were counted (random EM fields). Bar heights: Mean; error bars: s.e.m. (n = 3 mice/condition, two-way ANOVA with Tukey's multiple comparisons test, $F_{(1,8)}$ = 12.22, p=0.0081; $p_{control\ vs\ KO\ vehicle}$ = 0.0008, $p_{KO\ vehicle\ vs\ rapamycin}$ = 0.0037). (k) Western blot analysis, P5 sciatic nerve lysates from controls and mutants treated with vehicle or rapamycin (n = 4 mice/condition). Quantification: *Figure 1—figure supplement 1b*. *p<0.05, **p<0.01, ***p<0.001.
DOI: https://doi.org/10.7554/eLife.29241.003

The following figure supplements are available for figure 1:

**Figure supplement 1.** Quantification of western blots.
DOI: https://doi.org/10.7554/eLife.29241.004

**Figure supplement 2.** Deletion of TSC1 inhibits the onset of SC myelination by hyperactivating mTORC1.

*Figure 1 continued on next page*

Figure 1 continued

DOI: https://doi.org/10.7554/eLife.29241.005

was also apparent in this experimental setting (*Figure 2—figure supplement 2b*). Together, these results demonstrate that in SCs mTORC1 is mainly activated by the PI3K-Akt pathway, and that, in turn, high mTORC1 activity suppresses Akt activation without significantly affecting the MAPK pathway or activation of ErbB2-ErbB3 receptors.

To address whether the observed Akt suppression is responsible for the defective myelination due to TSC1 deletion in SCs, we sought to restore Akt activity in TSC1 mutants by simultaneously deleting the phospholipid phosphatase PTEN which limits activation of the PI3K-Akt pathway (*Song et al., 2012*) (*Figure 2b*). Given the known tumor suppressor role of both TSC1 and PTEN (*Inoki et al., 2005*), we used the more restricted $Mpz^{Cre}$ mouse line (*Feltri et al., 1999*) to generate suitable single or double mutants, referred to as $Mpz^{Cre}:Tsc1^{KO}$, $Mpz^{Cre}:Pten^{KO}$, and $Mpz^{Cre}:Tsc1^{KO}:Pten^{KO}$. Both proteins could be successfully depleted (*Figure 2—figure supplement 2c,d*). As anticipated, co-deletion of TSC1 and PTEN released the previously observed suppression on the PI3K-Akt pathway upon single deletion of TSC1, indicated by restored phosphorylation of Akt at T308 (*Figure 2c*, *Figure 2—figure supplement 1b*). We also noticed, based on the resulting levels of phospho-S6$^{S235/236}$, that $Mpz^{Cre}:Tsc1^{KO}:Pten^{KO}$ nerves contained higher mTORC1 activity compared to $Mpz^{Cre}:Tsc1^{KO}$ (*Figure 2c*, *Figure 2—figure supplement 1b*). To explain this finding, we hypothesized that the high Akt activity in $Mpz^{Cre}:Tsc1^{KO}:Pten^{KO}$ may further increase mTORC1 activity via phosphorylation and subsequent inhibition of PRAS40, an inhibitory protein associated with mTORC1 (*Sancak et al., 2007*) (*Figure 2b*). In agreement, $Mpz^{Cre}:Tsc1^{KO}:Pten^{KO}$ nerves displayed increased PRAS40 phosphorylation at the Akt-dependent phospho-site T246 compared to $Mpz^{Cre}:Tsc1^{KO}$ nerves (*Figure 2c*, *Figure 2—figure supplement 1b*). On the morphological level, P5 $Mpz^{Cre}:Tsc1^{KO}:Pten^{KO}$ nerves looked profoundly larger than those of controls or single mutants (*Figure 2d*). Nevertheless, EM analysis revealed no detectable improvement in SC differentiation compared to single mutants, but rather an aggravation of the phenotype with SCs in contact with sorted axons arrested at the promyelinating stage and virtually no myelinated fibers (*Figure 2e,f*). Unexpectedly, closer inspection of P5 $Mpz^{Cre}:Pten^{KO}$ mice revealed also a substantial decrease in myelinated fibers compared to controls due to SCs arrested at the promyelinating stage. Consistent with the morphological findings, cJun levels were comparably increased in both single knockout nerves compared to controls, and significantly more increased in double knockout nerves compared to single mutants (*Figure 2g*, *Figure 2—figure supplement 1c*).

These findings indicated that reduced Akt signaling due to mTORC1 hyperactivation in $Mpz^{Cre}:Tsc1^{KO}$ is likely not responsible for the observed impaired SC differentiation. On the other hand, the data also suggested that hyperactivation of the PI3K-Akt pathway early in development is per se detrimental to SC differentiation, potentially via mTORC1. In line with these conclusions, we found that SC-specific expression of constitutively active (myristoylated) Akt1, Akt2, or Akt3 in DRG-explant cultures inhibited myelination (*Figure 2h*, *Figure 2—figure supplement 2e*). To corroborate our findings and to determine the aforementioned potential role of high mTORC1 signaling in PTEN mutants, we generated PTEN knockout mice using $Dhh^{Cre}$-mediated recombination ($Dhh^{Cre}:Pten^{KO}$) and treated the resulting animals with vehicle or rapamycin (treatments at P3 and P4, analyses at P5). As observed in $Mpz^{Cre}:Pten^{KO}$ mice previously (*Figure 2f*), also vehicle-treated $Dhh^{Cre}:Pten^{KO}$ animals showed a decrease in myelinated fibers compared to controls, and rapamycin treatment effectively improved the $Dhh^{Cre}:Pten^{KO}$ phenotype (*Figure 2i,j*). In line with the morphological rescue, rapamycin also reduced expression of Oct6 and cJun (*Figure 2k*, *Figure 2—figure supplement 1d*).

## The PI3K-Akt-mTORC1 axis regulates Krox20 expression via S6K

To elucidate the molecular basis of how hyperactive mTORC1 impairs SC differentiation, we compared the transcriptomes of P5 $Dhh^{Cre}:Tsc1^{KO}$ and $Dhh^{Cre}:Pten^{KO}$ nerves, both displaying high mTORC1 activity, with age-matched nerves in which mTORC1 activity has been eliminated by conditional ablation of Raptor in SCs ($Dhh^{Cre}:Rptor^{KO}$) (*Norrmén et al., 2014*), together with controls. Unbiased hierarchical clustering showed that the replicates within each genotype group closely

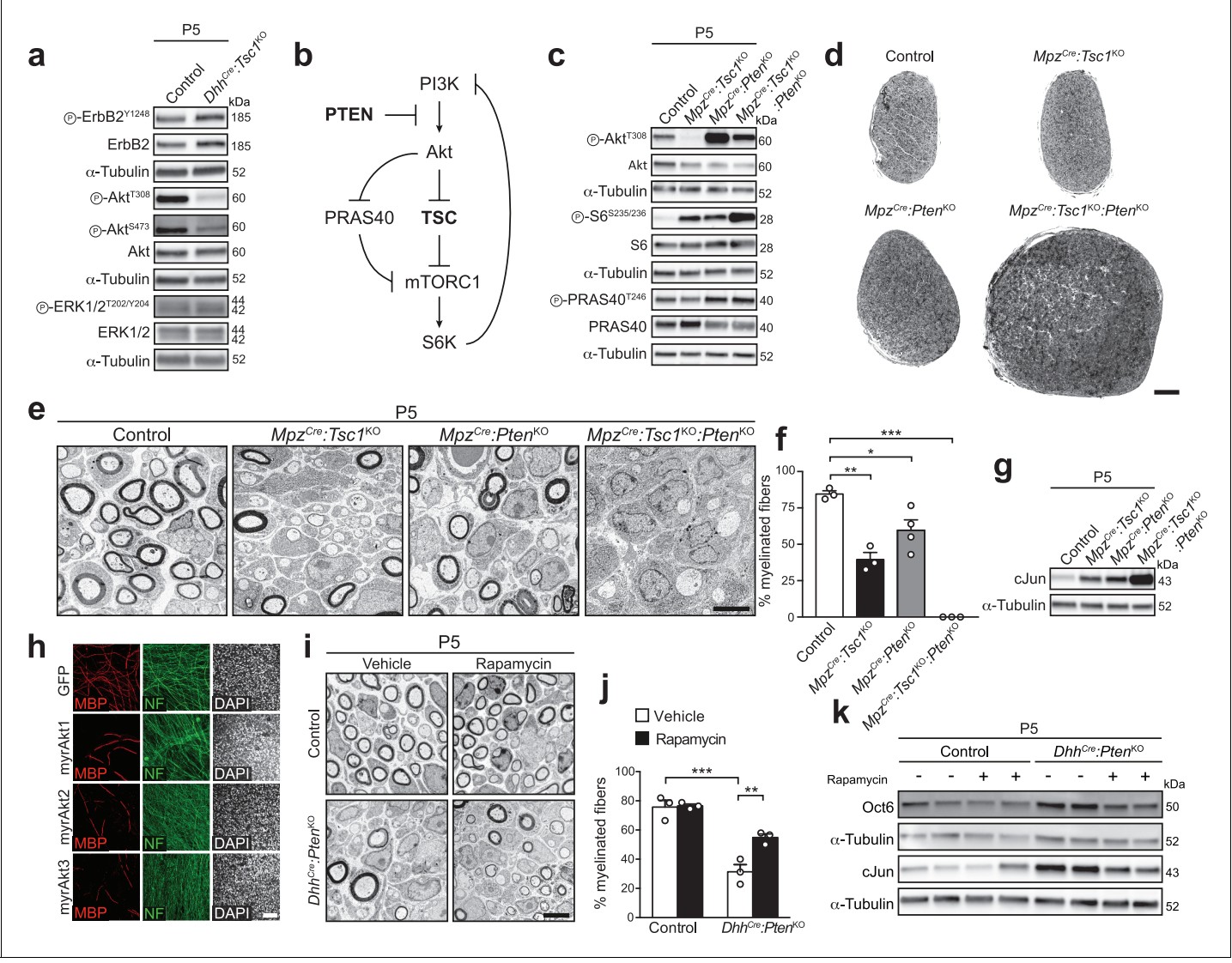

**Figure 2.** Deletion of PTEN restores Akt activity, but does not rescue the differentiation delay in TSC1-mutant mice. (**a**) Western blot analysis, P5 sciatic nerve lysates from control and $Dhh^{Cre}:Tsc1^{KO}$ mice (n = 3 mice/genotype). Quantification: *Figure 2—figure supplement 1a*. (**b**) Scheme of roles of PTEN and the TSC complex within the PI3K-Akt-mTORC1 axis. (**c**) Western blot analysis, P5 sciatic nerve lysates from control, $Mpz^{Cre}:Tsc1^{KO}$, $Mpz^{Cre}:Pten^{KO}$, and $Mpz^{Cre}:Tsc1^{KO}:Pten^{KO}$ mice (n = 3 mice/genotype). Quantification: *Figure 2—figure supplement 1b*. (**d**) Semithin cross sections of P5 control and mutant sciatic nerves (n = 3 mice/genotype). Scale bar: 100 μm. (**e**) Electron micrographs of P5 control and mutant sciatic nerves (n = 3 mice/genotype). Scale bar: 4 μm, refers to whole panel. (**f**) Myelinated fibers relative to all sorted fibers. Bar height: Mean; error bars: s.e.m. (n = 3 controls, TSC1-mutants, double-mutants and n = 4 PTEN-mutants, one-way ANOVA with Tukey's multiple comparisons test, $F_{(3,9)}$ = 44.98, p<0.0001; $p_{control\ vs\ TSC1KO}$ = 0.001, $p_{control\ vs\ PTENKO}$ = 0.028, $p_{control\ vs\ doubleKO}$ <0.0001). At least 150 sorted fibers per sample were counted (random EM fields). (**g**) Western blot analysis, P5 sciatic nerve lysates from controls and mutants (n = 3 mice/genotype). Quantification: *Figure 2—figure supplement 1c*. (**h**) Immunostainings of DRG-explant cultures infected with GFP- or Akt-expressing lentiviruses at day in vitro (DIV) 3. Quantification: *Figure 2—figure supplement 2e*. Three independent experiments performed, one representative experiment shown. MBP: myelin basic protein. NF: neurofilament. Scale bar: 100 μm, refers to whole panel. (**i**) Electron micrographs of P5 sciatic nerves from control and $Dhh^{Cre}:Pten^{KO}$ mice treated with vehicle or rapamycin (n = 3 mice/condition). Scale bar: 4 μm, refers to whole panel. (**j**) Quantification referring to i. Myelinated fibers relative to all sorted fibers. At least 150 sorted fibers per sample were counted (random EM fields). Bar height: Mean; error bars: s.e.m. (n = 3 mice/condition, two-way ANOVA with Tukey's multiple comparisons test, $F_{(1,8)}$ = 9.485, p=0.0151; $p_{control\ vs\ KO\ vehicle}$ = 0.0001, $p_{KO\ vehicle\ vs\ rapamycin}$ = 0.0085). (**k**) Western blot analysis, P5 sciatic nerve lysates from controls and mutants treated with vehicle or rapamycin (n = 4 mice/condition). Quantification: *Figure 2—figure supplement 1d*. *p<0.05, **p<0.01, ***p<0.001.

DOI: https://doi.org/10.7554/eLife.29241.006

The following figure supplements are available for figure 2:

**Figure supplement 1.** Quantification of western blots.

*Figure 2 continued on next page*

*Figure 2 continued*

DOI: https://doi.org/10.7554/eLife.29241.007

**Figure supplement 2.** Reciprocal interactions between the mTORC1 pathway and the PI3K-Akt pathway in SCs.

DOI: https://doi.org/10.7554/eLife.29241.008

clustered together, thus validating comparisons. Notably, *Dhh^Cre^:Tsc1^KO^* and *Dhh^Cre^:Pten^KO^* samples formed a distinct cluster compared to the other groups, confirming also at a global transcriptional level the previously described similarities (*Figure 3a*). Gene ontology (GO) analyses of *Dhh^Cre^:Tsc1^KO^* and *Dhh^Cre^:Pten^KO^* data sets revealed a particular upregulation of transcripts associated with cell proliferation compared to controls (*Figure 3—figure supplement 1a,b*), consistent with our previous observations (*Figure 1f,g*) and with other reports (*Goebbels et al., 2010*). In addition, downregulated lipid biosynthesis-associated mRNAs were overrepresented in *Dhh^Cre^:Rptor^KO^* compared to controls (*Figure 3—figure supplement 1c*), in line with our previous findings (*Norrmén et al., 2014*).

Given our focus on cell differentiation, we scrutinized our datasets to identify transcription factors (TFs) potentially regulated by mTORC1. To this end, we selected two subsets out of significantly changed mRNAs (fold change >1.2, FDR < 0.05): (1) Upregulated in both *Dhh^Cre^:Tsc1^KO^* and *Dhh^Cre^:Pten^KO^* (i.e. high mTORC1 activity), but downregulated in *Dhh^Cre^:Rptor^KO^* (i.e. no mTORC1 activity) or (2) Downregulated in *Dhh^Cre^:Tsc1^KO^* and *Dhh^Cre^:Pten^KO^*, but upregulated in *Dhh^Cre^:Rptor^KO^*. Applying these criteria, we identified 110 putative mTORC1-induced genes (subset 1) and 131 potential mTORC1-repressed genes (subset 2), among which 7 and 16 encoded TFs, respectively (*Figure 3b,c*). Among the mTORC1-repressed TFs emerged Krox20 (also known as Egr2), a crucial TF for the differentiation of myelinating SCs (*Svaren and Meijer, 2008*; *Topilko et al., 1994*). Thus, we reasoned that Krox20 may link mTORC1 activity to the onset of SC myelination. First, we confirmed the RNA-sequencing results with qRT-PCRs for Krox20 and other TFs with well-established roles in PNS myelination (cJun, Oct6, Brn2, Sox10, Sox2, Id2). Only Krox20 displayed a pattern consistent with stringent mTORC1-dependent regulation, being downregulated in both *Dhh^Cre^:Tsc1^KO^* and *Dhh^Cre^:Pten^KO^* nerves and mildly, but significantly upregulated in *Dhh^Cre^:Rptor^KO^* nerves at P5 and P8 (*Figure 3d*, *Figure 3—figure supplement 1d*). Furthermore, protein levels of Krox20 were also increased in Raptor mutants and decreased in TSC1 mutants (P5; *Figure 3e*, *Figure 3—figure supplement 2a*). Krox20 downregulation correlated with the extent of mTORC1 hyperactivation: Krox20 levels were comparably reduced in mice with single deletion of TSC1 or PTEN compared to controls and barely detectable in double knockout mice at P5 (*Figure 3f*, *Figure 3—figure supplement 2b*). In further support of the critical relationship between Krox20 and the PI3K-Akt-mTORC1 axis, we found that acute inhibition of mTORC1 either with rapamycin or torin-1, an ATP-competitive inhibitor of mTOR, strikingly increased basal Krox20 expression in cultured SCs, as did pharmacological inhibition of PI3K with LY294002 (*Figure 3g,h*). Suppression of mTORC1 or PI3K also enhanced cyclic-AMP induced upregulation of Krox20 (*Figure 3i*). We then examined the roles of the classical mTORC1 targets, 4EBPs and S6K, in regulating Krox20. Overexpression of a constitutively active version of 4EBP1 (i.e. cannot be phosphorylated by mTORC1) did not significantly affect Krox20 expression (*Figure 3j*). In contrast, inhibition of S6K with LYS6K2 moderately increased basal Krox20 expression and partially rescued the defective myelination in DRG-explant cultures from *Dhh^Cre^:Tsc1^KO^* animals (*Figure 3k,l*, *Figure 3—figure supplement 1e*).

In conclusion, this set of data provides converging evidence that Krox20 is an mTORC1-regulated transcript, and that S6K is part of the mechanisms linking mTORC1 activity to Krox20 expression. Our results indicate that high mTORC1 activity suppresses expression of Krox20, the main driver of SC myelination, thereby inhibiting onset of myelination.

## Developmentally high mTORC1 activity requires critical downregulation to allow onset of myelination

Our results so far show a peculiar inhibitory role of high mTORC1 signaling in SC differentiation. In this light, mTORC1 activity in physiological conditions would be expected to be high before onset of myelination and to decrease as SCs differentiate into myelinating SCs. Thus, we evaluated mTORC1 and Akt activities throughout mouse nerve development. Consistently, we found high activity of

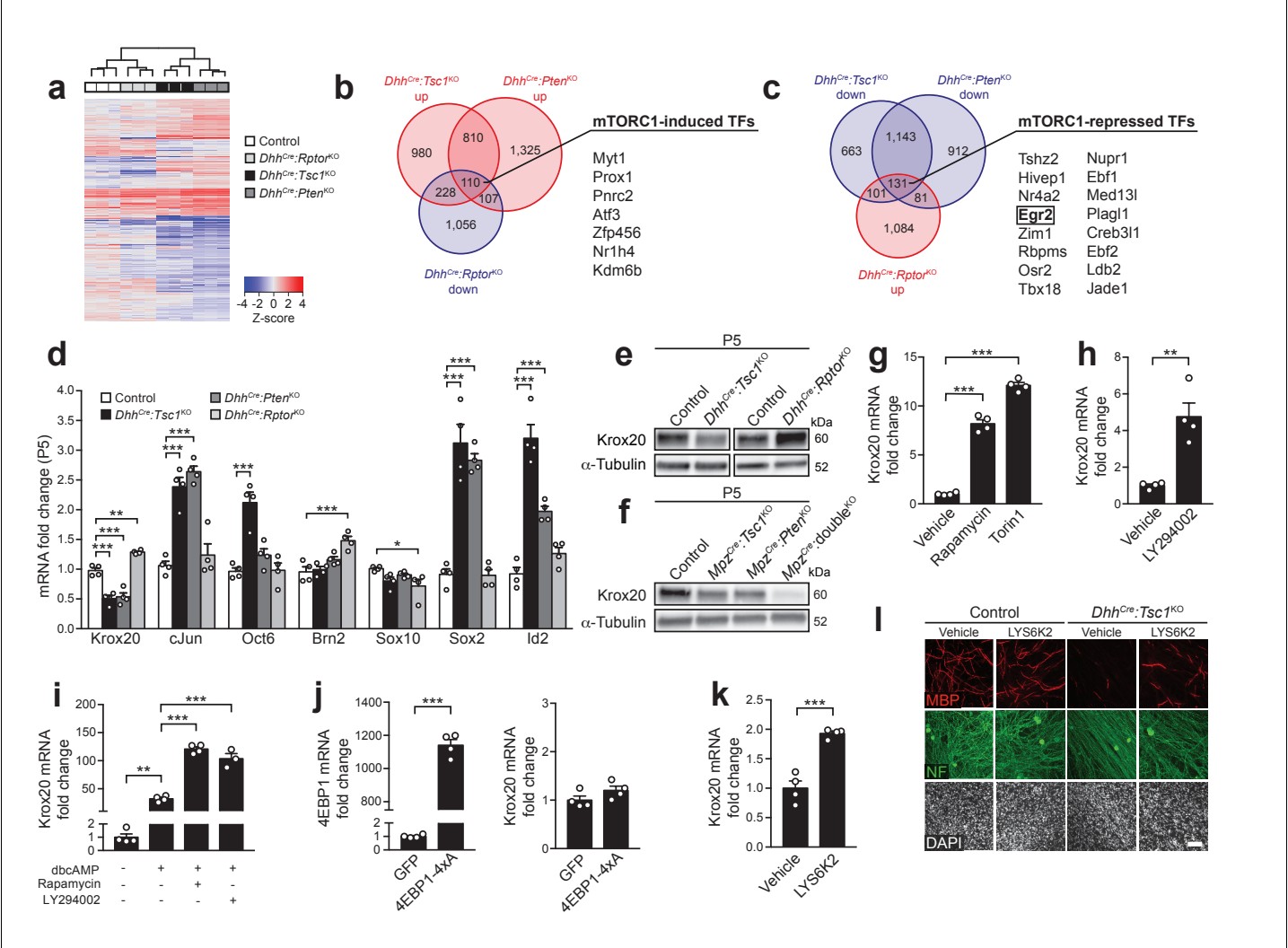

**Figure 3.** mTORC1 negatively regulates Krox20 expression via S6K. (**a**) Cluster analysis and heatmap of transcriptome profiles from P5 control, *Dhh^Cre*:*Rptor*^KO, *Dhh^Cre*:*Tsc1*^KO, and *Dhh^Cre*:*Pten*^KO sciatic nerves (n = 3 mice/genotype). (**b,c**) Venn diagrams of differentially expressed genes relative to controls (fold change >1.2, FDR < 0.05). (**d**) qRT-PCR analysis from P5 sciatic nerves. Transcript levels relative to control, after normalization to β-actin. Bar height: Mean; error bars: s.e.m. (n = 4 mice/genotype, one-way ANOVA with Dunnett's multiple comparisons test, $F_{(3,12)Krox20}$ = 61.75, $F_{(3,12)cJun}$ = 34.14, $F_{(3,12)Oct6}$ = 19.85, $F_{(3,12)Brn2}$ = 13.11, $F_{(3,12)Sox10}$ = 3.747, $F_{(3,12)Sox2}$ = 44.76, $F_{(3,12)Id2}$ = 49.23, $p_{Krox20}$ <0.0001, $p_{cJun}$ <0.0001, $p_{Oct6}$ <0.0001, $p_{Brn2}$ <0.0001, $p_{Sox10}$ = 0.0414, $p_{Sox2}$ <0.0001, $p_{Id2}$ <0.0001; Krox20, $p_{control\ vs\ TSC1KO}$ = 0.0001, $p_{control\ vs\ PTENKO}$ = 0.0001, $p_{control\ vs\ RaptorKO}$ = 0.0017; cJun, $p_{control\ vs\ TSC1KO}$ = 0.0001, $p_{control\ vs\ PTENKO}$ = 0.0001; Oct6, $p_{control\ vs\ TSC1KO}$ = 0.0001; Brn2, $p_{control\ vs\ RaptorKO}$ = 0.0003; Sox10, $p_{control\ vs\ RaptorKO}$ = 0.0212; Sox2, $p_{control\ vs\ TSC1KO}$ = 0.0001, $p_{control\ vs\ PTENKO}$ = 0.0001; Id2, $p_{control\ vs\ TSC1KO}$ = 0.0001, $p_{control\ vs\ PTENKO}$ = 0.0006). (**e,f**) Western blot analysis, P5 sciatic nerve lysates (n = 3 mice/genotype). Quantification: *Figure 3—figure supplement 2a,b*. (**g**) qRT-PCR from primary rat SCs in growth medium treated with vehicle (DMSO), rapamycin (20 nM), or torin-1 (20 nM) for 24 hr. Transcript levels relative to vehicle, after normalization to GAPDH. Bar height: Mean; error bars: s.e.m. (n = 4 biological replicates/condition, one-way ANOVA with Dunnett's multiple comparisons test, $F_{(2,9)}$ = 378, p<0.0001; $p_{vehicle\ vs\ rapamycin}$ = 0.0001, $p_{vehicle\ vs\ torin1}$ = 0.0001). Three independent experiments performed, one exemplary experiment shown. (**h**) qRT-PCR from primary rat SCs in growth medium treated with vehicle (DMSO) or LY294002 (20 μM) for 24 hr. Transcript levels relative to vehicle, after normalization to GAPDH. Bar height: Mean; error bars: s.e.m. (n = 4 biological replicates/condition, two-tailed unpaired Student's t-test, $t_{(6)}$ = 4.969, p=0.0025). Three independent experiments performed, one exemplary experiment shown. (**i**) qRT-PCR from primary rat SCs in defined medium with or without dbcAMP and treated with vehicle (DMSO), rapamycin (20 nM), or LY294002 (20 μM) for 24 hr. Transcript levels relative to defined medium-treated cells, after normalization to GAPDH. Bar height: Mean; error bars: s.e.m. (n = 3 biological replicates for LY294002 treated cells, n = 4 biological replicates for the other conditions, one-way ANOVA with Tukey's multiple comparisons test, $F_{(3,11)}$ = 179.5, p<0.0001; $p_{defined\ medium\ vs\ dbcAMP\ vehicle}$ = 0.0012, $p_{dbcAMP\ vehicle\ vs\ dbcAMP\ rapamycin}$ <0.0001, $p_{dbcAMP\ vehicle\ vs\ dbcAMP\ LY294002}$ <0.0001). Three independent experiments performed, one exemplary experiment shown. (**j**) qRT-PCR from primary rat SCs in growth medium 48 hr after transfection with plasmids expressing GFP or constitutively active 4EBP1 (4EBP1-4xA). Transcript levels relative to GFP-transfected cells, after normalization to GAPDH. Bar height: Mean; error bars: s.e.m. (n = 4 biological replicates/condition, two-tailed unpaired Student's t-test; 4EBP1, $t_{(6)}$ = 33.37, p<0.0001; Krox20, $t_{(6)}$ = 1.641, p=0.1519). Three independent experiments performed, one representative experiment shown. (**k**) qRT-PCR from primary rat SCs in growth medium treated with vehicle (DMSO) or LYS6K2 (3 μM)

*Figure 3 continued on next page*

*Figure 3 continued*

for 24 hr. Transcript levels relative to vehicle, after normalization to GAPDH. Bar height: Mean; error bars: s.e.m. (n = 4 biological replicates/condition, two-tailed unpaired Student's t-test, $t_{(6)}$ = 7.171, p=0.0004). Three independent experiments performed, one representative experiment shown. (I) Immunostaining of DRG-explant cultures from control or *Dhh^Cre:Tsc1*^KO embryos. Vehicle (DMSO) or LYS6K2 (3 μM) were added upon induction of myelination. Quantification: *Figure 3—figure supplement 1e*. Three independent experiments performed, one representative experiment shown. MBP: myelin basic protein. NF: neurofilament. Scale bar: 100 μm, refers to whole panel. *p<0.05, **p<0.01, ***p<0.001.
DOI: https://doi.org/10.7554/eLife.29241.009

The following source data and figure supplements are available for figure 3:

**Source data 1.** Transcriptomes of TSC1-, PTEN-, and Raptor-mutant nerves at P5 compared to controls.
DOI: https://doi.org/10.7554/eLife.29241.012
**Figure supplement 1.** Transcriptome profile of P5 sciatic nerves in various conditions of mTORC1 activation.
DOI: https://doi.org/10.7554/eLife.29241.010
**Figure supplement 2.** Quantification of western blots.
DOI: https://doi.org/10.7554/eLife.29241.011

mTORC1, reflected by the phosphorylation status of S6, from embryonic day 17.5 (E17.5) to P1, while Akt activation peaked at P1. In both cases, highest levels preceded detection of P0 and MBP (*Figure 4a,b*), in line with recent data obtained from postnatal rats (*Heller et al., 2014*). Conversely, as mTORC1 activity declined between E17.5 and P5, the levels of myelin proteins increased, as did the levels of Krox20 mRNA (*Figure 4c*), consistent with negative regulation of Krox20 expression by mTORC1. To examine mTORC1 activity in early nerve development at cellular resolution, we performed immunohistochemistry at P1. The majority of SCs highly expressing phospho-S6 were not yet myelinating (80.09 ± 2.35%, mean ±s.e.m., n = 4 mice) (*Figure 4d*, inset 1), while weaker phospho-S6 staining was found in association with myelinated fibers (*Figure 4d*, inset 2), consistent with previous cell culture data (*Heller et al., 2014*). High mTORC1 activity was often observed in arrangements reminiscent of axon bundles. These assemblies consist of SCs surrounding multiple axons and extending cytoplasmic processes to sort large caliber axons prior to myelination in a process called radial sorting. Consistently, the temporal span of high mTORC1 activity coincides with the period of intense radial sorting (*Figure 4a,b*). Radial sorting was impaired when mTORC1 was disrupted in *Dhh^Cre:Rptor*^KO nerves (*Norrmén et al., 2014*), indicating that the high mTORC1 activity observed in early nerve development is required in this process. Thus, we examined P5 *Mpz^Cre:Tsc1*^KO:*Pten*^KO nerves in which we had observed the highest mTORC1 activity (*Figure 2c*) for radial sorting alterations. We found that bundles often contained fewer axons than in control nerves (*Figure 4e*). Coherent with this finding, the number of sorted axons was significantly higher compared to controls (*Figure 4f*), indicative of increased radial sorting.

Collectively, we conclude that: (1) In normal nerve development, high mTORC1 activity is present before the onset of myelination and declines as SCs start myelinating; (2) The decrease in mTORC1 activity is physiologically required to allow SCs to differentiate into myelinating SCs, based on the findings that SC differentiation is impaired by sustained activation of mTORC1 in TSC1 and/or PTEN mutants; (3) High mTORC1 activity inhibits the differentiation of myelinating SCs, but promotes radial sorting, possibly to ensure that the differentiation program of myelination is activated only after radial sorting is completed.

## High mTORC1 signaling can reactivate radial myelin growth in adult SCs

According to our timeline analysis, the activity of the PI3K-Akt-mTORC1 axis in adult nerves is substantially lower than in early development (*Figure 4a,b*). Thus, we analyzed and compared systematically the effects of differently increased mTORC1 and/or PI3K-Akt signaling in adult SCs. To this end, we crossed our floxed mice with mice carrying a *Mpz^CreERT2* transgene, thus allowing inducible SC-specific ablation of TSC1 and/or PTEN (referred to as *Mpz^CreERT2:Tsc1*^KO, *Mpz^CreERT2:Pten*^KO, and *Mpz^CreERT2:Tsc1*^KO:*Pten*^KO). Tamoxifen was administered to young adult mice and three months later (months post-tamoxifen, mpt) TSC1 and/or PTEN protein levels were substantially reduced in the corresponding nerves (*Figure 5a–c*, *Figure 5—figure supplement 1a–c*). Western blot analyses for phospho-S6K^T389 and phospho-Akt^T308 showed that, like in development, deletion of TSC1 in adult nerves resulted also in hyperactivation of mTORC1 and suppression of Akt activation, while

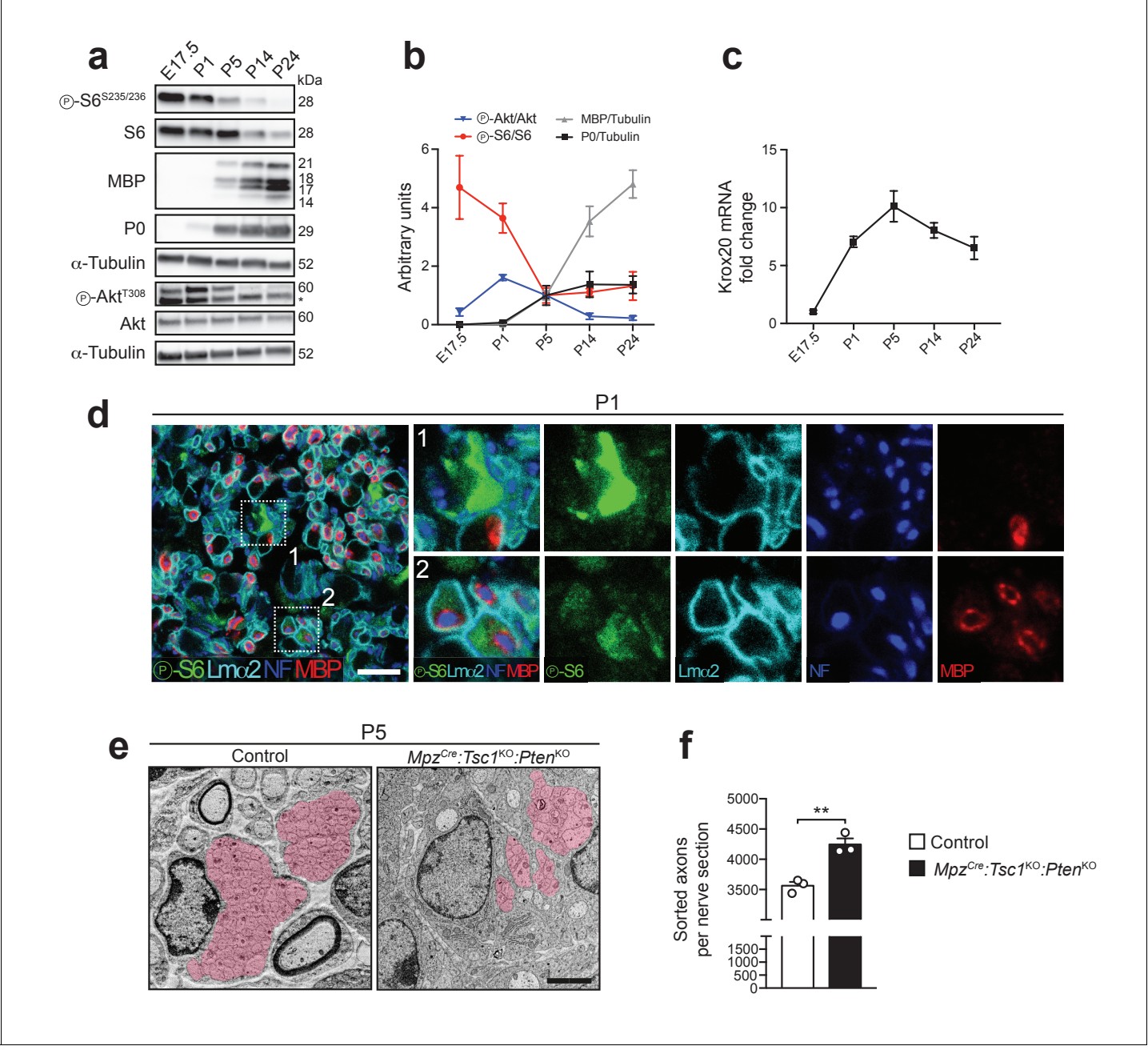

**Figure 4.** Akt and mTORC1 activities are high before onset of myelination and decline as myelination proceeds. (**a**) Western blot analysis of wild-type sciatic nerves (n = 3 mice/time point; for E17.5, three pools of sciatic nerves were used). Asterisk: Unspecific band. (**b**) Quantification referring to a, with P5 average set to 1. Symbols: Mean; error bars: s.e.m. (**c**) qRT-PCR from wild-type sciatic nerves. Transcript levels relative to E17.5, after normalization to α-tubulin. Symbols: Mean; error bars: s.e.m. (n = 6 mice at E17.5, n = 3 mice at P5, n = 4 mice for other time points). (**d**) Confocal microscopy images of immunostained sciatic nerve cross sections from P1 wild-type mice (n = 4 mice). Insets 1 and 2 show cells strongly or weakly positive for phospho-S6, in contact with axons, and surrounded by a basal lamina. Lmα2: laminin-α2. NF: neurofilament. MBP: myelin basic protein. Scale bar: 10 μm. (**e**) Electron micrographs of P5 control and $Mpz^{Cre}:Tsc1^{KO}:Pten^{KO}$ sciatic nerves. Area covered by bundles of unsorted axons highlighted in red (n = 3 mice/genotype). Scale bar: 2 μm, refers to whole panel. (**f**) Sorted axons per sciatic nerve cross section from P5 control and $Mpz^{Cre}:Tsc1^{KO}:Pten^{KO}$ mice. Bar height: Mean; error bars: s.e.m. (n = 3 mice/genotype, two-tailed unpaired Student's t-test, $t_{(4)}$ = 5.806, p=0.0044). \*\*p<0.01.

DOI: https://doi.org/10.7554/eLife.29241.013

The following source data is available for figure 4:

**Source data 1.** Timeline of PI3K-Akt-mTORC1 activity in mouse nerve development (data referring to Figure 4b)
DOI: https://doi.org/10.7554/eLife.29241.014

*Figure 4 continued*

**Source data 2.** Timeline of Krox20 expression in mouse nerve development (data referring to Figure 4c)

DOI: https://doi.org/10.7554/eLife.29241.015

deletion of PTEN or combined deletion of TSC1 and PTEN hyperactivated both PI3K-Akt and mTORC1 signaling. Notably, $Mpz^{CreERT2}:Tsc1^{KO}:Pten^{KO}$ mice displayed higher mTORC1 levels than either single knockout animals, reminiscent of development (**Figure 5d**, **Figure 5—figure supplement 1d**). Macroscopically, $Mpz^{CreERT2}:Pten^{KO}$ and $Mpz^{CreERT2}:Tsc1^{KO}:Pten^{KO}$ nerves were enlarged compared to controls and $Mpz^{CreERT2}:Tsc1^{KO}$ nerves at 3 mpt (**Figure 5e**). At the ultrastructural level, we found no signs of demyelination in controls or single mutants, while $Mpz^{CreERT2}:Tsc1^{KO}:Pten^{KO}$ nerves contained small numbers of demyelinated fibers (174 ± 45 per sciatic nerve cross section, mean ±s.e.m., n = 3 mice per genotype). Next, we evaluated myelin thickness expressed as g-ratio, i.e. the ratio between the axon diameter and the fiber diameter (axon plus myelin). All mutant nerves showed radial hypermyelination (i.e. increased myelin thickness without visible structural abnormalities) as indicated by lower g-ratios, with $Mpz^{CreERT2}:Tsc1^{KO}:Pten^{KO}$ animals being significantly more hypermyelinated than single knockouts (**Figure 5f,g**). In TSC1 mutants, radial hypermyelination was more pronounced in fibers with an axonal diameter smaller than 2 μm, whereas in $Mpz^{CreERT2}:Pten^{KO}$ and $Mpz^{CreERT2}:Tsc1^{KO}:Pten^{KO}$ nerves, axons of all sizes were significantly affected (**Figure 5h**). We did not observe substantial major shifts in axonal size distributions, pointing to a genuine increase in myelin thickness as reason for reduced g-ratios (**Figure 5i**). The observed radial hypermyelination was persistent at 6 mpt in $Mpz^{CreERT2}:Tsc1^{KO}$ and $Mpz^{CreERT2}:Pten^{KO}$ nerves (**Figure 5—figure supplement 2a–d**). Intriguingly, at 6 mpt, PTEN mutants were more hypermyelinated than TSC1 mutants and showed a stronger reduction in g-ratio compared to 3 mpt (**Figure 5—figure supplement 2b,f**).

In addition to radial hypermyelination, $Mpz^{CreERT2}:Pten^{KO}$ nerves exhibited various myelin abnormalities, including infoldings, outfoldings, and tomacula, both at 3 mpt and 6 mpt (**Figure 5f,j**, **Figure 5—figure supplement 2e**), consistent with a previous study (**Goebbels et al., 2012**). In $Mpz^{CreERT2}:Tsc1^{KO}$ mice such alterations were slightly, but not significantly, increased, while $Mpz^{CreERT2}:Tsc1^{KO}:Pten^{KO}$ nerves contained more myelin abnormalities than single mutants.

Finally, many Remak bundles in PTEN and double mutants displayed redundant SC membrane wrapping around non-myelinated axons, while such features were undetectable in controls or TSC1 knockouts (**Figure 5k**, **Figure 5—figure supplement 3a**). These findings are in line with previous reports in PTEN mutants (**Goebbels et al., 2010**), as is our confirmatory observation of occasional redundant membranes wrapped around collagen fibers in these mice (**Figure 5—figure supplement 3b**).

Together, our data show that high mTORC1 signaling following deletion of TSC1 and/or PTEN in adult differentiated SCs is capable of reactivating radial myelin growth. This effect is proportional to the levels of mTORC1 hyperactivation. Additionally, the subtly different phenotypes of TSC1 and PTEN mutants, including the presence of abnormal wrapping of Remak fibers in PTEN and double mutants, but not in TSC1 mutants, are consistent with potential mTORC1-independent functions of the PI3K-Akt pathway in membrane wrapping and myelin growth.

## The differentiation status of SCs determines the outcome of mTORC1 activation

In marked contrast to the outcome in early PNS development, hyperactivation of the PI3K-Akt-mTORC1 axis in adulthood promotes myelin growth. These two opposing effects point to a dual role of mTORC1 signaling in normal SC myelination: Inhibition of SC differentiation before the onset of myelination, but promotion of myelin production after myelination has started. This dual role model predicts that the response of SCs to mTORC1 hyperactivation depends on the SC differentiation status. Thus, we performed two additional sets of experiments, in which either SCs in adult nerves with hyperactive mTORC1 were forced to dedifferentiate and redifferentiate, or mTORC1 was hyperactivated in SCs that had just differentiated into myelinating SCs.

First, we induced SC dedifferentiation by carrying out nerve crush injuries in TSC1- or PTEN-conditional knockout backgrounds. Upon nerve injury, SCs dedifferentiate, engage with regrowing

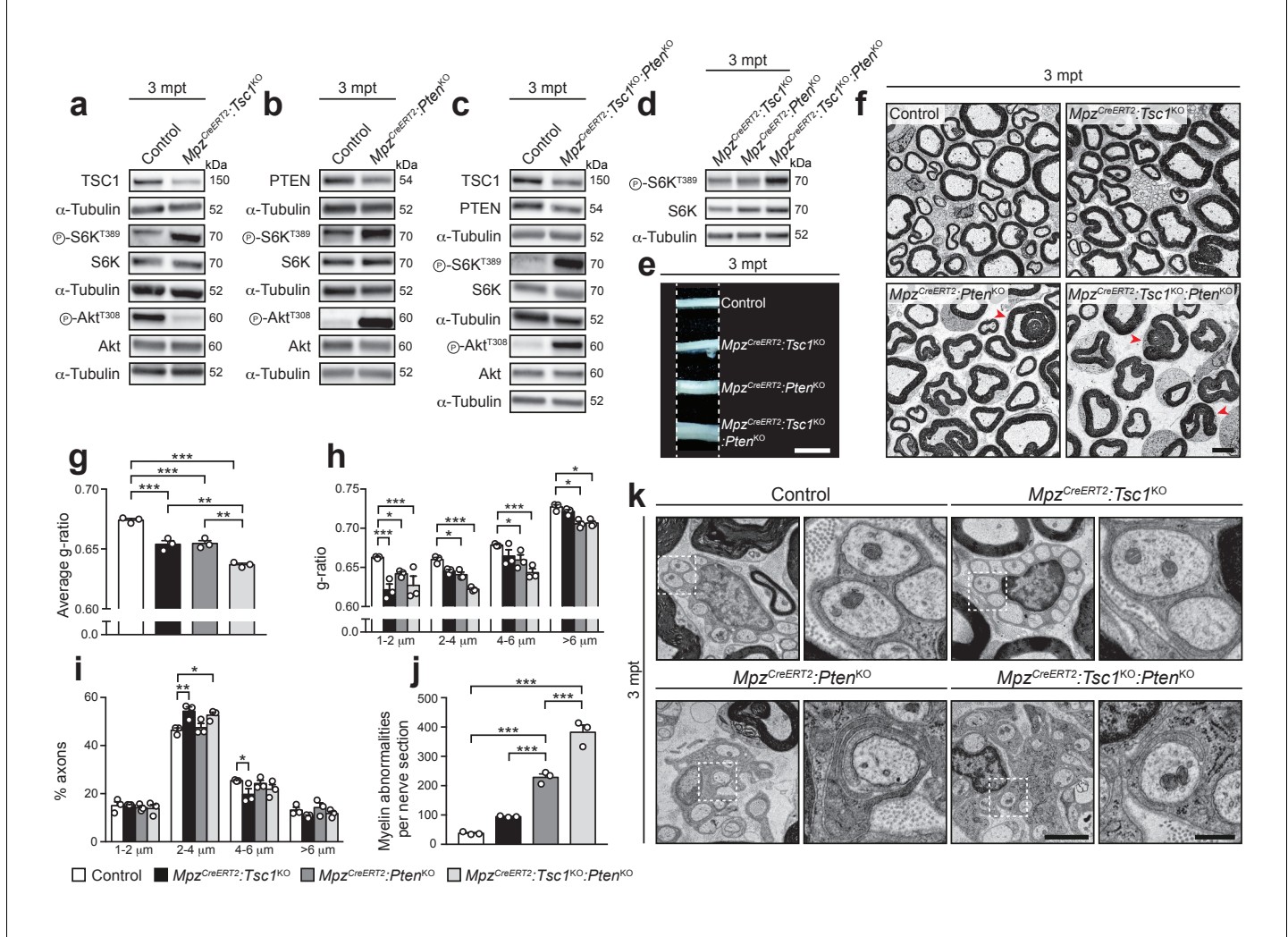

**Figure 5.** Inducible deletion of TSC1 and/or PTEN in adult mice reactivates myelin growth. (**a–d**) Western blot analysis, 3 mpt sciatic nerve lysates from control, *Mpz*<sup>*CreERT2*</sup>:*Tsc1*<sup>KO</sup>, *Mpz*<sup>*CreERT2*</sup>:*Pten*<sup>KO</sup>, and *Mpz*<sup>*CreERT2*</sup>:*Tsc1*<sup>KO</sup>:*Pten*<sup>KO</sup> mice (n = 3 mice/genotype). Quantification: *Figure 5—figure supplement 1a–d*. Note that exposure times for the three phospho-Akt$^{T308}$ western blots in a, b, and c were adjusted per membrane to avoid saturation of signals. (**e**) Macroscopic appearance of 3 mpt sciatic nerves from controls and mutants. Scale bar: 2.5 mm. (**f**) Electron micrographs of sciatic nerve cross sections from 3 mpt controls and mutants (n = 3 mice/genotype). Scale bar: 4 µm, refers to whole panel. (**g–i**) G-ratio, g-ratio distribution, and axon diameter distribution of 3 mpt control and mutant sciatic nerves. At least 400 sorted fibers per sample were analyzed (random EM fields). Bar height: Mean; error bars: s.e.m. (n = 3 mice/genotype; g, one-way ANOVA with Tukey's multiple comparisons test, $F_{(3,8)}$ = 52.91, p<0.0001, $p_{control\ vs\ TSC1KO}$ = 0.0006, $p_{control\ vs\ PTENKO}$ = 0.0008, $p_{control\ vs\ doubleKO}$ <0.0001, $p_{TSC1KO\ vs\ doubleKO}$ = 0.0018, $p_{PTENKO\ vs\ doubleKO}$ = 0.0014; h, two-way ANOVA with Tukey's multiple comparisons test, $F_{(9,32)}$ = 2.699, p=0.0185, $p_{1-2\ µm\ control\ vs\ TSC1KO}$ <0.0001, $p_{1-2\ µm\ control\ vs\ PTENKO}$ = 0.0269, $p_{1-2\ µm\ control\ vs\ doubleKO}$ <0.0001, $p_{2-4\ µm\ control\ vs\ PTENKO}$ = 0.0453, $p_{2-4\ µm\ control\ vs\ doubleKO}$ <0.0001, $p_{4-6\ µm\ control\ vs\ PTENKO}$ = 0.0485, $p_{4-6\ µm\ control\ vs\ doubleKO}$ = 0.0001, $p_{>6\ µm\ control\ vs\ PTENKO}$ = 0.0212, $p_{>6\ µm\ control\ vs\ doubleKO}$ = 0.0299; (i), two-way ANOVA with Tukey's multiple comparisons test, $F_{(9,32)}$ = 4.055, p=0.0015, $p_{2-4\ µm\ control\ vs\ TSC1KO}$ = 0.0022, $p_{2-4µm\ control\ vs\ doubleKO}$ = 0.0188, $p_{4-6\ µm\ control\ vs\ TSC1KO}$ = 0.0357). (**j**) Myelin abnormalities per sciatic nerve cross sections at 3 mpt. Bar height: Mean; error bars: s.e.m. (n = 3 mice/genotype, one-way ANOVA with Tukey's multiple comparisons test, $F_{(3,8)}$ = 123.3, p<0.0001; $p_{control\ vs\ PTENKO}$ <0.0001, $p_{control\ vs\ doubleKO}$ <0.0001, $p_{PTENKO\ vs\ TSC1KO}$ = 0.0006, $p_{PTENKO\ vs\ doubleKO}$ = 0.0002). (**k**) Electron micrographs of 3 mpt sciatic nerves showing exemplary Remak bundles and enlarged details (dashed square). (n = 3 mice/genotype). Scale bar: 2 µm (left), 500 nm (right). *p<0.05, **p<0.01, ***p<0.001.

DOI: https://doi.org/10.7554/eLife.29241.016

The following figure supplements are available for figure 5:

**Figure supplement 1.** Quantification of western blots.
DOI: https://doi.org/10.7554/eLife.29241.017

**Figure supplement 2.** Radial hypermyelination in inducible knockout mice for TSC1 or PTEN is persistent at 6 mpt.
DOI: https://doi.org/10.7554/eLife.29241.018

*Figure 5 continued on next page*

Figure 5 continued

**Figure supplement 3.** Redundant membrane wrapping in inducible knockout mice for PTEN, alone or in combination with TSC1.
DOI: https://doi.org/10.7554/eLife.29241.019

axons, and remyelinate them in a highly regulated sequence of events that recapitulates many aspects of developmental myelination (*Chen et al., 2007*; *Scherer et al., 1994*; *Zorick et al., 1996*), but also involves a special repair program (*Jessen and Mirsky, 2016*). We reasoned that, if our model is valid, inducible deletion of TSC1 or PTEN should lead to impaired remyelination when mice undergo a nerve crush injury. Nerve surgeries were performed in $Mpz^{CreERT2}$:$Tsc1^{KO}$ and $Mpz^{CreERT2}$:$Pten^{KO}$ animals at 2 mpt and nerves were analyzed morphologically at 5, 12, and 30 days post-crush (dpc) (*Figure 6a*). Quantification of intact-appearing myelin profiles (defined as non-discontinuous and non-collapsed myelin rings) at 5 dpc revealed no major differences in demyelination between mutants and controls (*Figure 6b,c*) allowing comparative analysis of subsequent remyelination (although a subtle influence of the maintenance phenotypes of $Mpz^{CreERT2}$:$Tsc1^{KO}$ and $Mpz^{CreERT2}$:$Pten^{KO}$ mice cannot be excluded). At 12 dpc, when initial remyelination occurs, both $Mpz^{CreERT2}$:$Tsc1^{KO}$ and $Mpz^{CreERT2}$:$Pten^{KO}$ nerves showed a marked reduction in remyelinated fibers. In TSC1 knockouts, a minor reduction was persistent at 30 dpc, when remyelination initiation in controls was virtually completed (*Figure 6b,d*). In line with defective remyelination initiation, the percentage of SCs expressing Krox20 was reduced at 12 dpc in $Mpz^{CreERT2}$:$Pten^{KO}$ mice, while the overall proportions of SCs were not changed (*Figure 6e–g*). Consistently, Oct6 levels were increased in $Mpz^{CreERT2}$:$Tsc1^{KO}$ nerves at the same time point (*Figure 6h*, *Figure 6—figure supplement 1a*).

In the second set of experiments, we genetically hyperactivated mTORC1 during developmental myelination, but only after most SCs have started myelinating. According to our model, constitutive activation of mTORC1 in cells that have just differentiated to myelinating SCs should lead to radial hypermyelination. To achieve this goal, we crossed $Tsc1$-floxed mice with mice carrying a $Plp1^{CreERT2}$ transgene (yielding $Plp1^{CreERT2}$:$Tsc1^{KO}$). We induced recombination by administering 4-hydroxyta-moxifen (4OH-TMX) (*Zuchero et al., 2015*) from P8 to P12 since most SCs have started myelinating at this time (*Arroyo et al., 1998*), and compared these inducible mutant mice with the previously described $Dhh^{Cre}$:$Tsc1^{KO}$ mice, in which recombination occurs before onset of myelination (*Figure 6i*), together with controls. $Plp1^{CreERT2}$-driven deletion of TSC1 was successful and led to constitutive mTORC1 activation, as indicated by increased phosphorylation of S6K at T389 (*Figure 6j*, *Figure 6—figure supplement 1b*). Consistent with our model, P60 $Plp1^{CreERT2}$:$Tsc1^{KO}$ nerves were radially hypermyelinated, in contrast to the hypomyelinated $Dhh^{Cre}$:$Tsc1^{KO}$ nerves (*Figure 6k,l*).

## Discussion

Our results reveal that PI3K-Akt-mTORC1 signaling directs PNS myelination at multiple critical levels, including onset of myelination and myelin growth. We started our studies by conditionally deleting TSC1 to explore the effects of hyperactivating mTORC1 in developing SCs. Unexpectedly, the resulting constitutively elevated mTORC1 activity profoundly delayed SC differentiation to myelinating cells. Searching for an explanation, we investigated whether inhibitory feedback of mTORC1 on PI3K-Akt signaling, shown in other cases to account for paradoxical results upon hyperactivation of mTORC1 (*Byles et al., 2013*; *Tang et al., 2014*; *Yecies et al., 2011*), might be responsible. However, the phenotype of TSC1 mutants was even aggravated after restoring Akt activity in SCs via simultaneous PTEN deletion. Moreover, we found that single deletion of PTEN leading to high Akt activity in vivo, or expression of constitutively active Akt in cell cultures, inhibited the onset of myelination per se. Notably as in TSC1 mutants, inhibition of SC differentiation by hyperactive PI3K-Akt due to loss of PTEN was mediated by mTORC1 since rapamycin treatment effectively rescued this effect.

These data unveiled a direct myelination-inhibitory function of the PI3K-Akt-mTORC1 axis. In support of this paradigm, we discovered that mTORC1 negatively regulates Krox20, the main TF required for onset of SC myelination, via S6K. Coherently, we found that mTORC1 activity is physiologically high in not-yet myelinating SCs and declines with their commitment to myelination. In this light, the impaired onset of myelination in TSC1 and/or PTEN mutants can be viewed as a failure of

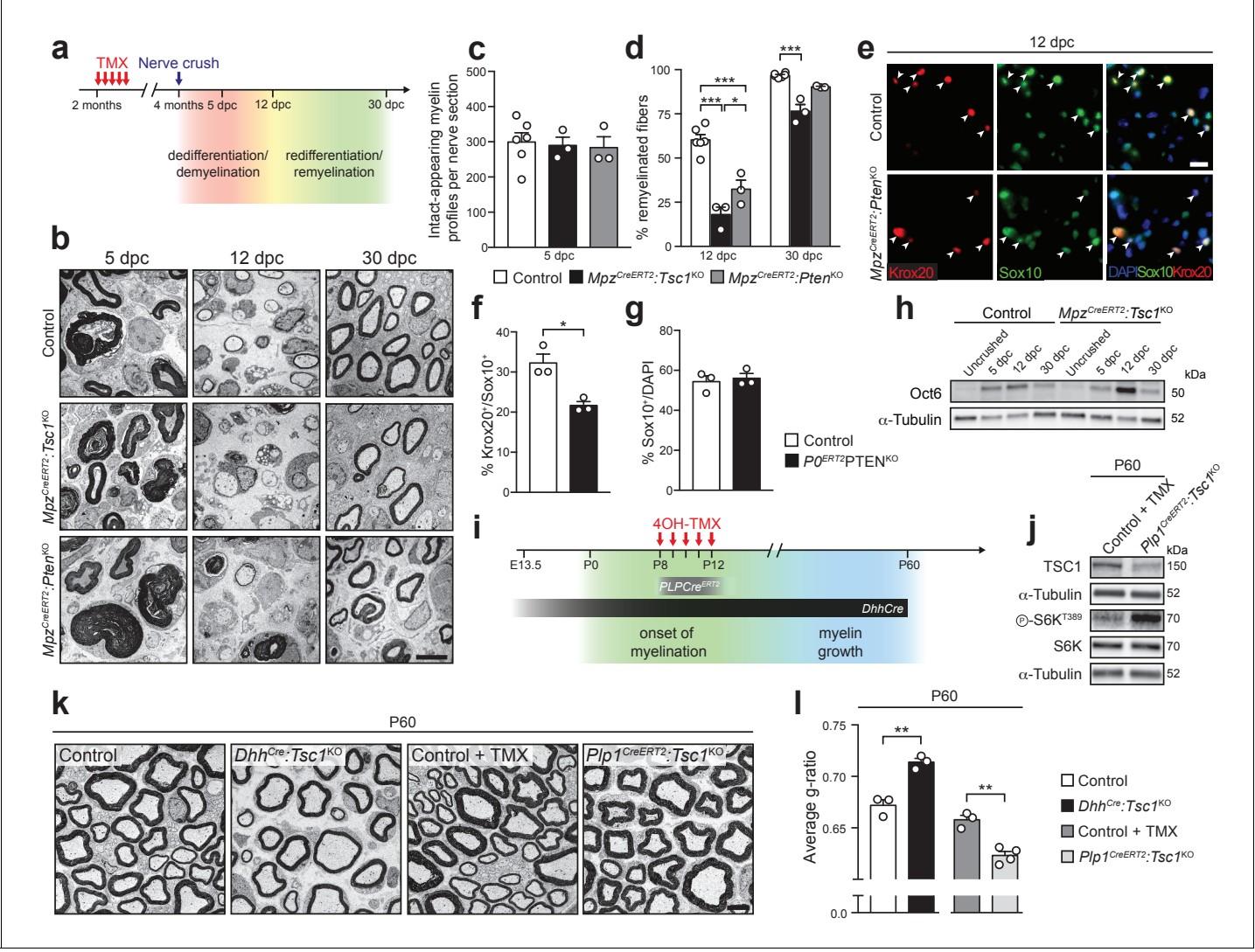

**Figure 6.** The differentiation status of SCs determines the outcome of mTORC1 hyperactivation. (**a**) Nerve crush injury outline. TMX: Tamoxifen. (**b**) Electron micrographs of control, $Mpz^{CreERT2}$:$Tsc1$^KO, and $Mpz^{CreERT2}$:$Pten$^KO sciatic nerves at 5, 12, and 30 dpc (n = 6 control mice and 3 mice for each mutant group/time point). Scale bar: 5 μm, refers to whole panel. (**c**) Intact-appearing myelin profiles per sciatic nerve at 5 dpc. Bar height: Mean; error bars: s.e.m. (n = 6 control mice and 3 mice for each mutant group, one-way ANOVA, $F_{(2,9)}$ = 0.08361, p=0.9205). (**d**) Remyelinated fibers relative to all sorted fibers. At least 100 sorted fibers per sample were counted (random EM fields). Bar height: mean; error bars: s.e.m. (n = 6 control mice and 3 mice for each mutant group, two-way ANOVA with Tukey's multiple comparisons test, $F_{(2,18)}$ = 10.37, p=0.0010; 12 dpc, $p_{control\ vs\ TSC1KO}$<0.0001, $p_{control\ vs\ PTENKO}$ <0.0001, $p_{TSC1KO\ vs\ PTENKO}$ = 0.0196; 30 dpc, $p_{control\ vs\ TSC1KO}$ = 0.0003). (**e**) Immunostaining of sciatic nerves at 12 dpc (n = 3 mice/genotype). Nuclei: DAPI. Arrowheads: Krox20/Sox10 double-positive nuclei. Scale bar: 10 μm, refers to whole panel. (**f,g**) Quantification referring to e. Krox20/Sox10 double-positive of Sox10-positive nuclei (**f**), percentage of Sox10-positive nuclei (**g**). Bar height: Mean; error bars: s.e.m. (n = 3 mice/genotype, two-tailed unpaired Student's t-test; f, $t_{(4)}$ = 4.366, p=0.012; g, $t_{(4)}$ = 0.4453, p=0.6792). (**h**) Western blot analysis of lysates of contralateral and crushed sciatic nerves (n = 3 mice/genotype for every time point). Quantification: *Figure 6—figure supplement 1a*. (**i**) Scheme depicting Cre activity in $Dhh^{Cre+}$ animals or $Plp1^{CreERT2+}$ animals treated with 4-hydroxytamoxifen (4OH-TMX) from P8 to P12. (**j**) Western blot analysis, P60 sciatic nerve lysates from 4-hydroxytamoxifen injected control (Control +TMX) and $Plp1^{CreERT2}$:$Tsc1$^KO animals (n = 3 mice/genotype). Quantification: *Figure 6—figure supplement 1b*. (**k**) Electron micrographs of P60 sciatic nerve cross sections from control, $Dhh^{Cre}$:$Tsc1$^KO, Control +TMX and $Plp1^{CreERT2}$:$Tsc1$^KO animals (n = 4 $Plp1^{CreERT2}$:$Tsc1$^KO animals, n = 3 for other conditions). Scale bar: 4 μm, refers to whole panel. (**l**) G-ratio analysis referring to k. At least 200 sorted fibers per sample were analyzed (random EM fields). Bar height: Mean; error bars: s.e.m. (n = 4 $Plp1^{CreERT2}$:$Tsc1$^KO animals, n = 3 for the other conditions, two-tailed unpaired Student's t-test, $t_{(4)control\ vs\ DhhCre:Tsc1KO}$ = 6.346, $t_{(5)control+TMX\ vs\ Plp1CreERT2:Tsc1KO}$ = 5.917, $p_{control\ vs\ DhhCre:Tsc1KO}$ = 0.0032, $p_{control\ +TMX\ vs\ Plp1CreERT2:Tsc1KO}$ = 0.002). *p<0.05, **p<0.01, ***p<0.001.
DOI: https://doi.org/10.7554/eLife.29241.020

The following figure supplement is available for figure 6:

**Figure supplement 1.** Quantification of western blots.

*Figure 6 continued on next page*

*Figure 6 continued*

DOI: https://doi.org/10.7554/eLife.29241.021

mutant SCs to promptly downregulate mTORC1 activity when myelination has to start. What is then the physiological function of high mTORC1 during early nerve development? We have shown previously that genetic inhibition of mTORC1 impairs radial sorting (*Norrmén et al., 2014*) and describe now that strong hyperactivation of mTORC1 can accelerate this crucial process in myelination. We infer that high mTORC1 activity during early nerve development may drive radial sorting, concomitant with transiently inhibiting further SC differentiation to myelinating cells, thereby serving as a key component of the regulatory program that coordinates completion of radial sorting and onset of myelination (*Feltri et al., 2016*).

Intriguingly, recent studies on melanocytes, which share a common developmental origin with SCs, revealed parallels between mTORC1 functions in these cell types. Analogous to the differentiation-inhibiting role of mTORC1 in SCs and the negative regulation of Krox20 by mTORC1, mTORC1 hyperactivation in melanocytes impaired melanogenesis, while inhibition of mTORC1 increased levels of MITF, a TF essential for melanin production (*Cao et al., 2017*; *Ho et al., 2011*). Hence, it will be appealing to explore whether an inhibitory role of mTORC1 activity on cell differentiation is a frequent feature of neural crest-derived cells.

Our results show that physiologically high activity of the PI3K-Akt-mTORC1 axis before the ordinary developmental onset of myelination has a myelination-inhibitory function. However, it has been previously reported that conditional ablation of PTEN in the Schwann cell lineage increased the number of myelinated fibers in 3 months-old mice, most likely due to aberrant myelination of normally non-myelinated C fibers (*Goebbels et al., 2010*). Thus, chronic hyperactivation of PI3K-Akt signaling may be able to, in the long term, drive Remak cells to a myelinating-like phenotype (see also below).

In contrast to the inhibitory effect on SC differentiation, we found that deletion of TSC1 and/or PTEN in adult SCs was able to reactivate myelin growth, proportionally to the levels of mTORC1 activity. However, in spite of comparable mTORC1 hyperactivity, we noticed subtle phenotypical differences between $Mpz^{CreERT2}:Tsc1^{KO}$ and $Mpz^{CreERT2}:Pten^{KO}$ nerves. First, $Mpz^{CreERT2}:Pten^{KO}$, but not $Mpz^{CreERT2}:Tsc1^{KO}$, displayed increased radial hypermyelination at 6 mpt compared to 3 mpt, indicating that specifically PTEN ablation led to continuous radial myelin growth. Second, in $Mpz^{CreERT2}:Pten^{KO}$ and $Mpz^{CreERT2}:Tsc1^{KO}:Pten^{KO}$, but not in $Mpz^{CreERT2}:Tsc1^{KO}$, redundant processes of non-myelinating SCs repeatedly wrapped around axons in Remak bundles, analogous to previous observations (*Domènech-Estévez et al., 2016*; *Goebbels et al., 2010*). These findings indicate that the PI3K-Akt pathway is likely to serve also mTORC1-independent functions in controlling SC myelination. In particular, the redundant wrapping of SC membranes upon loss of PTEN, but not of TSC1, prompts us to speculate that, also during physiological myelination, driving of SC membrane wrapping might involve PI3K-Akt-dependent/mTORC1-independent mechanisms. We envisage that the PI3K-Akt pathway serves converging scopes during myelin growth, recruiting (1) mTORC1 to activate the synthesis of myelin building blocks, and (2) mTORC1-independent targets for membrane wrapping. The existence of such a PI3K-Akt-dependent/mTORC1-independent 'wrapping force' in myelination is supported by recent reports (*Domènech-Estévez et al., 2016*; *Mathews and Appel, 2016*) and may underlie the regulation of the mechanistically required cytoskeletal dynamics. In agreement, recent work demonstrated that constitutively active Akt induces Rac1 activity in SCs (*Domènech-Estévez et al., 2016*), and remodeling of the actin cytoskeleton is an essential prerequisite for membrane expansion during myelination (*Nawaz et al., 2015*; *Novak et al., 2011*; *Zuchero et al., 2015*).

The formation of myelin abnormalities – a hallmark of numerous neuropathies (*Dyck and Thomas, 2005*) – may also involve mTORC1-independent targets of the PI3K-Akt pathway, given that such alterations accumulated substantially in $Mpz^{CreERT2}:Pten^{KO}$, but less so in $Mpz^{CreERT2}:Tsc1^{KO}$. Nevertheless, high mTORC1 signaling probably supports their growth, since myelin abnormalities were more abundant in $Mpz^{CreERT2}:Tsc1^{KO}:Pten^{KO}$ with higher mTORC1 activity compared to single $Mpz^{CreERT2}:Pten^{KO}$. In accord, treatment with rapamycin has been shown to diminish the load of myelin abnormalities in PTEN mutant mice (*Goebbels et al., 2012*).

Taken together, also supported by data from others (*Beirowski et al., 2017*), our results support a model in which the PI3K-Akt-mTORC1 axis fulfills multiple major roles in SC myelination (*Figure 7*). In early SC development, high mTORC1 signaling maintains SCs in a non-differentiated state by suppressing Krox20 expression and, at the same time, promotes radial sorting. After axonal sorting is completed, a physiological decline in mTORC1 activity releases the suppression on Krox20 expression and allows myelination to start. When this turning point has been passed, the residual mTORC1 activity drives myelin production in concert with mTORC1-independent targets of PI3K-Akt. Downstream of mTORC1, both lipid synthesis via activation of the RXRγ-SREBP1c axis (*Norrmén et al., 2014*) and protein synthesis (*Sheean et al., 2014*) are key processes supporting myelin growth.

Our model is especially supported by the opposing outcomes of mTORC1 hyperactivation in developing non-differentiated SCs versus in adult myelinating SCs, revealing that SCs reacted differently depending on their differentiation status. To confirm this notion, we forced adult differentiated SCs to dedifferentiate due to nerve crush injuries, followed by axonal regeneration and SC redifferentiation. As expected in this experimental setting, redifferentiation and remyelination were delayed

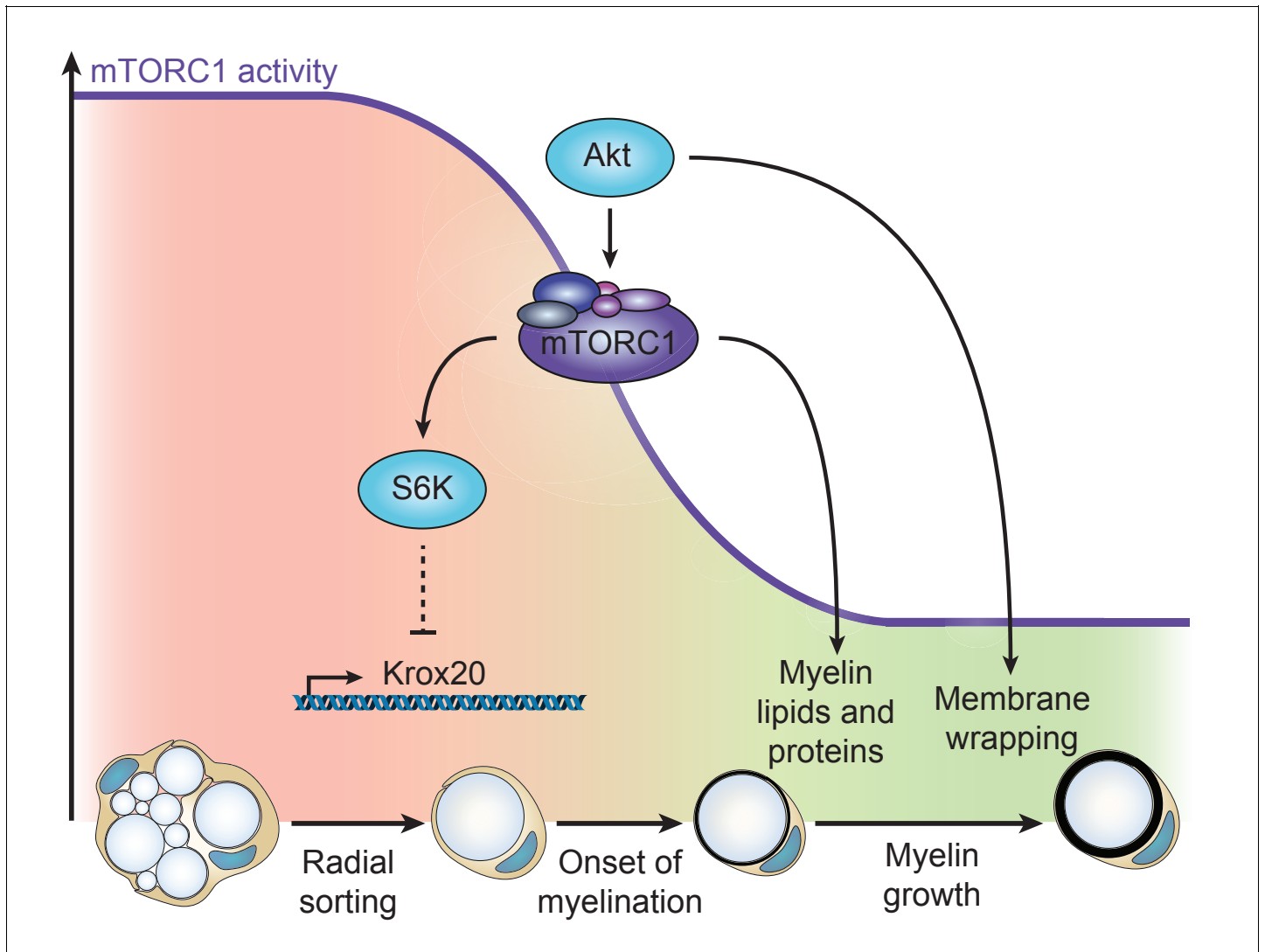

**Figure 7.** Model of the dual role of the PI3K-Akt-mTORC1 axis in SC myelination. Before onset of myelination, high activity of the pathway inhibits the differentiation of SCs by negatively regulating Krox20 transcription via S6K. After SCs have started myelinating, mTORC1 and Akt synergize to promote myelin growth: mTORC1 enhances lipid synthesis, while Akt, additionally to activating mTORC1, is likely to drive membrane wrapping largely independent of mTORC1.

DOI: https://doi.org/10.7554/eLife.29241.022

in $Mpz^{CreERT2}$:$Tsc1^{KO}$ and $Mpz^{CreERT2}$:$Pten^{KO}$ mutants. Conversely, deletion of TSC1 when most SCs had just started myelinating enhanced radial myelin growth.

Our conceptual model raises a number of questions for future investigations including: Are different upstream receptors mediating the differentiation-inhibiting and myelin-growth promoting roles of mTORC1? Is a single upstream receptor system accounting for both, but through different ligands (e.g. neuregulin-1 isoforms), or through different concentrations of the same ligand? What is responsible for the physiological decline in mTORC1 activity that allows SCs to start myelinating? What are the postulated mTORC1-independent targets of PI3K-Akt? We can only speculate in this context, but we have shown here that neuregulin-1 signaling is a major activator of the PI3K-Akt-mTORC1 axis in SCs, in line with previously reported findings of strongly reduced phospho-S6 levels in SCs co-cultured with DRG neurons missing neuregulin-1 (*Heller et al., 2014*). Neuregulin-1 isoforms are known to exert different effects on SCs in various experimental settings involving several signaling pathways (*Mei and Nave, 2014*). Thus, signaling through different neuregulin-1 isoforms might contribute to the different functions of mTORC1 in SC biology, potentially in concert with additional signals integrating other adaxonal and also abaxonal cues (*Ghidinelli et al., 2017*; *Heller et al., 2014*; *Herbert and Monk, 2017*; *Monk et al., 2015*; *Pereira et al., 2012*).

Modulation of the pathways upstream of mTORC1 has been recently explored as a promising therapeutic approach in animal models of hereditary peripheral neuropathies (*Bolino et al., 2016*; *Fledrich et al., 2014*; *Goebbels et al., 2012*; *Nicks et al., 2014*). Based on our 'dual-role' model, we expect that future studies will benefit from monitoring mTORC1 activity in conjunction with the differentiation status of SCs to enhance efficacy. Moreover, inhibition of mTORC1 should be considered as a valuable therapeutic strategy in hereditary or acquired peripheral neuropathies in which SC differentiation is defective.

## Materials and methods

### Animal procedures

Mice harboring floxed alleles of *Tsc1* (STOCK Tsc1<tm1Djk>/J, RRID:IMSR_JAX:005680) and *Pten* (C;129S4-Pten<tm1Hwu>/J, RRID:IMSR_JAX:004597) were obtained from The Jackson Laboratory. Mice harboring floxed alleles of *Rptor* (*Bentzinger et al., 2008*; *Polak et al., 2008*) and mice carrying a Cre transgene under control of the *Dhh* (RRID:IMSR_JAX:012929) or *Mpz* promoter (RRID: IMSR_JAX:017927), or a CreERT2 transgene under control of the *Plp1* or *Mpz* promoters have been described (*Feltri et al., 1999*; *Jaegle et al., 2003*; *Leone et al., 2003*). To generate non-inducible conditional deletion of TSC1, PTEN, or Raptor, floxed mice were crossed with $Dhh^{Cre}$- or $Mpz^{Cre}$-positive mice. To generate inducible conditional deletion of TSC1 or PTEN, floxed mice were crossed with $Mpz^{CreERT2}$-positive mice, and at 8–10 weeks of age 2 mg of tamoxifen (Sigma-Aldrich, Saint Louis, MO, USA) in 10% ethanol/sunflower seed oil (Sigma-Aldrich) were injected intraperitoneally once a day on five consecutive days in both mutant and control animals. Developmental inducible deletion of TSC1 was achieved crossing floxed mice with $Plp1^{CreERT2}$-positive mice and administering intraperitoneally to both mutant and control pups 75 µg per gram of body weight of 4-hydroxytamoxifen (Sigma-Aldrich) dissolved in Kolliphor EL (Sigma-Aldrich) once a day on five consecutive days from P8 to P12. Experimental animals were on a hybrid background between C57B6 and the background of origin of the floxed mice (2–4 backcrosses with C57B6 for all TSC1, PTEN, or TSC1/PTEN mutants and their controls; 7–8 backcrosses with C57B6 for Raptor mutants and their controls). Cre-negative animals (floxed homozygous or heterozygous) were used as controls in the experiments. To control for variations in background, littermate controls were used. Wild-type mice were on a C57B6 background. Mice of either sex were used in the experiments. For in vivo rescues, rapamycin (Millipore, Billerica, MA, USA) was dissolved in a solution of 5% PEG-400, 5% Tween-80, and 4% ethanol, and 5 µg rapamycin per gram of body weight were administered intraperitoneally once a day at P3 and P4. Genotypes were determined through genomic PCR using the following primers: Cre forward 5'-accaggttcgttcactcatgg-3', reverse 5'-aggctaagtgcctcttctaca-3'; TSC1 forward 5'-gtcacgaccgtaggagaagc-3', reverse 5'-gaatcaaccccacagagcat-3'; PTEN forward 5'-caagcactctgc-gaactgag-3', reverse 5'-aagtttttgaaggcaagatgc-3', Raptor forward 5'-atggtagcaggcacactcttcatg-3', reverse 5'-gctaaacattcagtccctaatc-3'. Mice were housed with a maximum number of five animals per cage, kept in a 12 hr light-dark cycle, and fed standard chow *ad libitum*.

## Surgical procedures

Mice were subjected to unilateral sciatic nerve crush injury two months after tamoxifen administration. After inducing anesthesia with isoflurane inhalation, the sciatic nerve was exposed through blunt dissection of the thigh muscles and compressed for 30 s. For analgesia 0.1 mg kg$^{-1}$ buprenorphine (Temgesic, Reckitt Benckiser, UK) were injected intraperitoneally once prior to the surgery, and thereafter every 12 hr for two days. Morphological analysis and immunostaining of crushed nerves were performed on sections 3 mm distal to the injury site. For biochemical analysis, the whole nerve distal to the injury site was used.

## Morphological analysis and g-ratio measurements

Immediately after dissection, sciatic nerves were fixed with 3% glutaraldehyde and 4% paraformaldehyde in 0.1 M phosphate buffer. Sciatic nerves were further treated with 2% osmium tetroxide (EMS, Hatfield, PA, USA), dehydrated over a series of acetone gradients and embedded in Spurrs resin (EMS). Semithin sections (650 nm) were stained with 1% toluidine blue and used for qualitative analysis and for quantifying intact myelin profiles after nerve crush. Ultrathin sections (65 nm) were imaged with a FEI Morgagni 268 TEM, and random 5 × 5 MIA (multiple image alignment) fields were acquired for qualitative analysis and for quantifying the percentage of myelinating SCs. G-ratio measurements and quantification of myelin abnormalities were performed on EM reconstructions of the entire sciatic nerve, obtained from additional sections (99 nm) collected on ITO coverslips (Optics Balzers, Germany) and imaged with either a Zeiss Gemini Leo 1530 FEG or Zeiss Merlin FEG scanning electron microscopes attached to ATLAS modules (Zeiss, Germany). To calculate the g-ratio, the axon diameter was derived from the axon area, while the fiber diameter was calculated adding to the axon diameter twice the average of the myelin thickness measured at two different locations of the myelin ring. At least 200 fibers per nerve were analyzed.

## Antibodies and chemical inhibitors

The following primary antibodies were used: TSC1 (CST, #6935, 1:1,000), phospho-S6K$^{T389}$ (Cell Signaling Technology Cat# 9234 also 9234L, 9234P, 9234S RRID:AB_2269803, 1:1,000), S6K (Cell Signaling Technology Cat# 9202S RRID:AB_331676, 1:1,000), phospho-4EBP1$^{T37/46}$ (Cell Signaling Technology Cat# 2855 also 2855L, 2855S, 2855P RRID:AB_560835, 1:1,000), 4EBP1 (Cell Signaling Technology Cat# 9452S RRID:AB_10693791, 1:1,000), phospho-S6$^{S235/236}$ (Cell Signaling Technology Cat# 4857S RRID:AB_2181035, 1:1,000), S6 (Cell Signaling Technology Cat# 2317S RRID:AB_10694551, 1:1,000), cJun (Cell Signaling Technology Cat# 9165 RRID:AB_2130165, 1:1,000), phospho-ErbB2$^{Y1248}$ (Cell Signaling Technology Cat# 2247S RRID:AB_331725, 1:1,000), ErbB2 (Cell Signaling Technology Cat# 2165S RRID:AB_10692490, 1:1,000), phospho-Akt$^{T308}$ (Cell Signaling Technology Cat# 9275 also 9275S, 9275L RRID:AB_329828 only for the western blot in *Figure 4a*, Cell Signaling Technology Cat# 4056 also 4056S, 4056L RRID:AB_331163 for the other western blots, both used 1:1,000), phospho-Akt$^{S473}$ (Cell Signaling Technology Cat# 4051 also 4051S, 4051L RRID:AB_331158, 1:1,000), Akt (Cell Signaling Technology Cat# 9272 also 9272S RRID:AB_329827, 1:1,000), phospho-ERK1/2 $^{T202/Y204}$ (Cell Signaling Technology Cat# 9106 also 9106L, 9106S RRID:AB_331768, 1:1,000), ERK1/2 (Cell Signaling Technology Cat# 9102 also 9102L, 9102S RRID:AB_330744, 1:1,000), phospho-PRAS40$^{T246}$ (CST, #13175, 1:1,000), PRAS40 (Cell Signaling Technology Cat# 2691S RRID:AB_2225033, 1:1,000), PTEN (Cell Signaling Technology Cat# 9559S RRID:AB_10695541, 1:1,000), Akt phospho-substrates (Cell Signaling Technology Cat# 10001S RRID:AB_10950819, 1:1,000), α-tubulin (Sigma-Aldrich Cat# T5168 RRID:AB_477579, 1:1,000), β-actin (Sigma-Aldrich Cat# A5316 RRID:AB_476743, 1:1,000), GAPDH (Hytest Cat# 5G4-9B3 RRID:AB_1616725, 1:10,000), NF (Sigma-Aldrich Cat# N5264 RRID:AB_477278, used for immunocytochemistry, 1:200), NF (Aves Labs Cat# NF-M RRID:AB_2313554, used for immunohistochemistry, 1:1,000), MBP (Bio-Rad/AbD Serotec Cat# MCA409S RRID:AB_325004, 1:200), Sox10 (R&D Systems, AF2864, 1:200), S100 (Dako Cat# Z0311 RRID:AB_10013383, 1:200), laminin-α2 (Sigma-Aldrich Cat# L0663 RRID:AB_477153, 1:200). Primary antibodies against Oct6 and Krox20 (both used 1:1,000) were generous gifts from Dr. Dies Mejer. HRP-, AP-, and fluorophore-conjugated secondary antibodies were purchased from Jackson ImmunoResearch and used 1:200 for immunostainings or 1:10,000 for western blots. Alexa488-coupled phalloidin (used 1:100) was from Life Technologies (Carlsbad, CA, USA) (RRID: AB_2315147).

The following chemical inhibitors were used in in vitro experiments: rapamycin (Sigma-Aldrich), LY294002 (Sigma-Aldrich), CI-1040 (Sigma-Aldrich), torin-1 (Tocris, UK), LYS6K2 (Focus Biomolecules, Plymouth Meeting, PA, USA).

## Western blots

Immediately after dissection, sciatic nerves were placed in ice-cold PBS, the epineurium was removed with fine forceps, and the nerves were snap-frozen in liquid nitrogen and stored at −80°C until further processing. To prepare lysates, nerves were ground on dry ice, mixed with PN2 lysis buffer (25 mM Tris-HCl pH 7.4, 95 mM NaCl, 10 mM EDTA, 2% SDS, protease and phosphatase inhibitors (Roche, Switzerland)), boiled, and spun for 15 min. 10–30 μg of proteins per sample were mixed 1:4 with sample buffer (200 mM Tris-HCl pH6.8, 40% glycerol, 8% SDS, 20% β-mercaptoethanol, 0.4% bromophenol blue), run on 4–15% polyacrylamide gradient gels (Biorad, Hercules, CA, USA), and blotted onto PVDF membranes (Millipore). After blocking with 5% milk in TBS-T, membranes were incubated overnight with primary antibodies diluted in 5% BSA in TBS-T. Chemiluminescent signals were generated using HRP- or AP-conjugated secondary antibodies and ECL (GE Healthcare), ECL prime (GE Healthcare), or CDP-Star (Roche), and detected using Fusion FX7 (Vilber Lourmat, Germany). If required, after signal detection membranes were stripped with a buffer containing β-mercaptoethanol (1.52 g Tris, 2 g SDS, 700 μl β-mercaptoethanol in 100 ml water, pH 6.8) for 15 min at 55°C and reprobed with antibodies. Quantification of band intensities was performed with ImageJ (version 1.50i). α-tubulin, β-actin, or GAPDH were used as normalization controls. For cropped blots, the control bands are displayed immediately below the other bands that belong to the same membrane. Full-length blots are shown in *Supplementary files 1–8*. Size markers refer to All Blue Precision Protein Standards (Biorad). Protein size is expressed as apparent molecular weight in kDa.

## Immunostaining

Immediately after dissection, sciatic nerves were fixed for one hour in 4% paraformaldehyde, cryo-preserved in 10% sucrose for one hour, 20% sucrose overnight, and then embedded in OCT (Tissue-Tek, Torrance, CA, USA). 8 μm thick cryosections were cut, blocked with blocking buffer (1% BSA, 10% goat or donkey serum, 0.1% triton X-100 in PBS) for one hour, incubated overnight with primary antibodies, incubated for one hour with fluorophore-conjugated secondary antibodies, counterstained with DAPI (Life Technologies), and mounted with Vectashield (Vector Laboratories, Burlingame, CA, USA). The same procedure was used for cell culture stainings. For cell proliferation experiments, mouse pups were injected with 50 μg EdU (Life Technologies) per gram of body weight and sacrificed one hour later. EdU staining was performed using the Click-iT EdU Alexa647 kit (Life Technologies) as per manufacturer's instructions. Immunostainings were imaged using an epifluorescence microscope (Zeiss Axio Imager.M2) equipped with a monochromatic CCD camera (sCMOS, pco.edge), or with a confocal microscope (Leica TCS SP8). One representative section per sample was imaged and analyzed.

## RNA extraction and qPCR analysis

Immediately after dissection, sciatic nerves were placed in ice-cold PBS, the epineurium was removed with fine forceps, and the nerves were snap-frozen in liquid nitrogen and stored at −80°C until needed. Total RNA was extracted using RNeasy kit (Qiagen, Germany) for developmental samples or Qiazol (Qiagen) for adult samples and cells as per manufacturer's instructions. 50–200 ng of total RNA were reverse transcribed using Maxima First Strand cDNA Synthesis Kit (Thermo Fisher Scientific, Waltham, MA, USA) as per manufacturer's instructions. qPCR reactions were performed using FastStart Essential DNA Green Master (Roche) and Light Cycler 480 II (Roche). The sequences of the primers used and their specificity are the following: Krox20 (mouse and rat) forward 5'-acagcctctacccggtggaagac-3', reverse 5'-cagagatgggagcgaagctactcggata-3'; cJun (mouse and rat) forward 5'-gccaagaactcggaccttctcacgtc-3', reverse 5'-tgatgtgcccattgctggactggatg-3'; Oct6 (mouse) forward 5'-gagcactcggacgaggatg-3', reverse 5'-cacgttaccgtagagggtgc-3'; Brn2 (mouse) forward 5'-tcaaatgccctaagccctcg-3', reverse 5'-cgggaggggtcatcctttc-3'; Sox10 (mouse) forward 5'-ccgaccagtaccctcacct-3', reverse 5'-tcaatgaaggggcgcttgt-3'; Sox2 (mouse) forward 5'-ggaaagggttcttgctgggt-3', reverse 5'-acgaaaacggtcttgccagt-3'; Id2 (mouse) forward 5'-catcagcatcctgtccttgc-3', reverse 5'-

ttctcctggtgaaatggctgat-3'; GAPDH (mouse and rat) forward 5'-ggtgaaggtcggtgtgaacggatttgg-3', reverse 5'-ggtcaatgaaggggtcgttgatggcaac-3'; β-actin (mouse) forward 5'-gtccacacccgccacc-3', reverse 5'-ggcctcgtcacccacatag-3'; α-tubulin (mouse) forward 5'-tcttagttgtcgggaacggt-3', reverse 5'-ggagatgcactcacgcatgata-3'. Relative mRNA fold changes for each gene were obtained by using the $2^{-\Delta\Delta Ct}$ method after normalization to GAPDH, β-actin, or α-tubulin.

## Plasmids and cloning

To generate lentiviral vectors for overexpression in SCs, the hPGK promoter of the pCCLsin.PPT. hPGK.PRE lentiviral backbone was replaced by a 1.1 kb fragment of the rat P0 promoter, as previously described (Norrmén et al., 2014). myrAkt constructs were obtained from Addgene (#9008, #9016, #9017) (Ramaswamy et al., 1999), PCR-amplified, and inserted between the AgeI and SalI restriction sites of the modified pCCLsin.PPT.hPGK.PRE vector. The 4EBP1-4xA construct was obtained from Addgene (#38240) (Thoreen et al., 2012) and subcloned between the BamHI and NheI restriction sites of the pcDNA3.1 vector. As control, the eGFP coding sequence from the pCCLsin.PPT.hPGK.PRE vector was subcloned between the NheI and EcoRI restriction sites of the pcDNA3.1 vector. All constructs were sequence-verified prior to usage.

## Preparation, culture, and use of primary SCs

To prepare embryonic SCs, E13.5 mouse DRGs were isolated, digested with trypsin-EDTA 0.25% (Life Technologies) for 30 min, resuspended in N2-medium (Advanced DMEM:F12 plus N2 supplement (Life Technologies)) with 50 ng ml$^{-1}$ 7S NGF (Millipore), and plated on uncoated dishes. After one week, the SC-neuron network was mechanically detached from the underlying fibroblast layer, digested with 0.25% trypsin (Sigma-Aldrich, T9201) and 0.1% collagenase (Sigma-Aldrich, C0130), and replated on PLL-coated dishes in N2-medium devoid of NGF. After reaching confluency, cells were trypsinized, centrifuged, and resuspended in flow buffer (PBS plus 2% FCS (Life Technologies)). Flow cytometry experiments were performed with LSR Fortessa (BD Biosciences, Franklin Lakes, NJ, USA) using between $10^5$ and $10^6$ cells per genotype, and the data were analyzed with the FlowJo software (RRID:SCR_008520, version 10.0.7). To account for differences in the number of recorded events for each sample, the flow cytometry profiles were normalized to their modes.

To prepare neonatal mouse SCs, sciatic nerves were dissected from P2 mouse pups, the epineurium was removed, and the nerves were digested with 1.25 mg ml$^{-1}$ trypsin (Sigma-Aldrich, T9201) and 2 mg ml$^{-1}$ collagenase (Sigma-Aldrich, C0130) for one hour. Cells were then centrifuged, resuspended in D-medium (DMEM-Glutamax plus 10% FCS (Life Technologies)), and seeded on PLL-coated 12 mm glass coverslips. After overnight incubation, cells were fixed with 4% paraformaldehyde and stained.

To prepare rat SCs, sciatic nerves were dissected from P2 Sprague-Dawley rat pups, the epineurium was removed, and nerves were digested with 1.25 mg ml$^{-1}$ trypsin (Sigma-Aldrich, T9201) and 2 mg ml$^{-1}$ collagenase (Sigma-Aldrich, C0130) for one hour. After centrifugation, the cells were resuspended in D-medium (DMEM-Glutamax plus 10% FCS (Life Technologies)) and plated on PLL-coated dishes. Contaminating fibroblasts were eliminated with 10 μM Ara-C treatment (Sigma-Aldrich) for 48 hr, followed by complement-mediated removal of fibroblasts using an anti-Thy1.1 antibody (Bio-Rad/AbD Serotec Cat# MCA04G RRID:AB_322809, 1:50). After a 10 min incubation with the anti-Thy1.1 antibody, rabbit complement (Millipore) was added for 40 min. SCs were then further expanded in SC growth medium (DMEM-Glutamax (Life Technologies), 10% FCS (Life Technologies), 5 μg ml$^{-1}$ bovine pituitary extract (Life Technologies), 2 μM forskolin (Sigma-Aldrich)), tested for mycoplasma contamination, aliquoted, and stored in liquid nitrogen. For differentiation experiments, SCs were transferred to defined medium (100 μg ml$^{-1}$ of holotransferrin (Sigma-Aldrich), 60 ng ml$^{-1}$ progesterone (Sigma-Aldrich), 1 μg ml$^{-1}$ insulin (Sigma-Aldrich), 16 μg ml$^{-1}$ putrescine (Sigma-Aldrich), 400 ng ml$^{-1}$ L-thyroxin (Sigma-Aldrich), 160 ng ml$^{-1}$ sodium selenite (Sigma-Aldrich), 10 ng ml$^{-1}$ triiodothyronine (Sigma-Aldrich), 38 ng ml$^{-1}$ dexamethasone (Sigma-Aldrich), and 300 μg ml$^{-1}$ BSA (Sigma-Aldrich) in DMEM:F12-Glutamax (Life Technologies)), and differentiation was induced with 1 mM dbcAMP (Sigma-Aldrich) for 24 hr. For experiments with neuregulin stimulation, SCs were serum-starved for 6.5–8 hr in DMEM-Glutamax with 0.05% FCS (Life Technologies) and subsequently treated with 10 ng ml$^{-1}$ recombinant human EGF domain of neuregulin-1 β1 (R&D Systems, Minneapolis, MN, USA). For all other experiments, SCs were kept in SC

growth medium. Transfection of rat SCs was performed using Lipofectamine 3000 (Life Technologies) as per manufacturer's instructions. Only SCs lower than passage five were used for experiments.

## Preparation, culture, and analysis of DRG explants

DRGs were isolated from E13.5 embryos, digested for 45 min with trypsin-EDTA 0.25% (Life Technologies), and seeded on 12 mm coverslips coated with 10% Matrigel (Corning, NY, USA) at a density of 1.2 DRGs per coverslip. DRGs from embryos with the same genotype were pooled before seeding. Cultures were kept in supplemented NB-medium (Neurobasal (Life Technologies), B27 supplement (Life Technologies), 4 g l$^{-1}$ D-glucose (Sigma-Aldrich), 2 mM L-glutamine (Life Technologies), 50 ng ml$^{-1}$ 7S NGF (Millipore)) for the first five days and then in C-medium (MEM-Glutamax (Life Technologies), 10% FCS (Life Technologies), 4 g l$^{-1}$ D-glucose (Sigma-Aldrich), 50 ng ml$^{-1}$ 7S NGF (Millipore)) supplemented with 50 μg ml$^{-1}$ ascorbic acid (Sigma-Aldrich) to induce myelination for an additional 8–10 days. Both supplemented NB- and C-medium were changed every other day. Before staining, DRG-cultures were fixed in 4% paraformaldehyde for 15 min and subsequently permeabilized with ice-cold methanol for 10 min. Images of whole coverslips were acquired using an epifluorescence microscope (Zeiss Axio Imager.M2) equipped with a monochromatic CCD camera (sCMOS, pco.edge) and an automated stage. Between 3 and 6 coverslips per condition were imaged and analyzed. To quantify the extent of myelination, four random fields per coverslip were selected, the MBP-positive and the NF-positive areas per field were measured with ImageJ (version 1.50i) after thresholding, and the MBP area/NF area ratio per coverslip was calculated as average of the MBP area/NF area ratios of the different fields. The same threshold was applied to all samples and conditions.

## Production and use of lentiviruses

HEK293T cells were from ATCC (Manassas, VA, USA), were not further authenticated, and were regularly monitored to assure lack of mycoplasma contamination. Subconfluent HEK293T cells were transfected in 10 cm dishes with lentiviral vectors and the packaging plasmids psPAX2 and pCMV-VSV-G using Lipofectamine 2000 as per manufacturer's instructions (Life Technologies). Non-concentrated viruses were collected 48 hr after transfection, aliquoted, and stored at −80°C until needed. For infection of DRG-explant cultures, non-concentrated viruses were mixed 1:1 with supplemented NB-medium and added to the cultures for 24 hr from DIV 3 to 4.

## RNA-sequencing

The quantity and quality of isolated RNA was determined with a Qubit (1.0) Fluorometer (Life Technologies) and a Bioanalyzer 2100 (Agilent, Santa Clara, CA, USA). The TruSeq Stranded mRNA Sample Prep Kit (Illumina, San Diego, CA, USA) was used in the succeeding steps. Briefly, total RNA samples (150 ng) were ribosome depleted and then reverse-transcribed into double-stranded cDNA with actinomycin added during first-strand synthesis. The cDNA samples were fragmented, end-repaired and polyadenylated before ligation of TruSeq adapters. The adapters contain the index for multiplexing. Fragments containing TruSeq adapters on both ends were selectively enriched with PCR. The quality and quantity of the enriched libraries were validated using Qubit (1.0) Fluorometer and the Bioanalyzer 2100 (Agilent). The product is a smear with an average fragment size of approximately 360 bp. The libraries were normalized to 10 nM in Tris-Cl 10 mM, pH 8.5 with 0.1% Tween-20. The TruSeq SR Cluster Kit v4-cBot-HS or TruSeq PE Cluster Kit v4-cBot-HS (Illumina) was used for cluster generation using 8 pM of pooled normalized libraries on the cBOT. Sequencing was performed on the Illumina HiSeq 2500 paired-end at 2 × 126 bp or single-end 126 bp using the TruSeq SBS Kit v4-HS (Illumina).

The raw reads were first cleaned by removing adapter sequences, trimming low quality ends, and filtering reads with low quality (phred quality <20) using Trimmomatic (*Bolger et al., 2014*). Sequence alignment of the resulting high-quality reads to the *Mus musculus* reference genome (build GRCm38) and quantification of gene level expression was carried out using RSEM (version 1.2.22) (*Li and Dewey, 2011*). To detect differentially expressed genes we applied count based negative binomial model implemented in the software package EdgeR (R version: 3.2.2, edgeR_3.12.0) (*Robinson et al., 2010*). The differential expression was assessed using an exact test adapted for

over-dispersed data. Genes showing altered expression (fold change >1.2) with adjusted (Benjamini and Hochberg method) p-value<0.05 (indicated as false discovery rate, FDR) were considered differentially expressed. Within this set of genes, downregulated and upregulated genes were separately subjected to gene ontology analysis of biological processes using the online tool Enrichr (http://amp.pharm.mssm.edu/Enrichr/).

## Statistical analysis

Data processing and statistical analyses were performed using GraphPad Prism (RRID:SCR_002798, version 7.0a) and Microsoft Excel (version 15.27). Data distribution was assumed to be normal and variances were assumed to be equal, although this was not formally tested due to low n number. Sample sizes were chosen according to sample sizes generally employed in the field. The investigators were blinded to the genotypes during analysis of morphological and immunohistochemical data, except for those cases in which mutant mice exhibited an obvious phenotype. No randomization methods were used. Two-tailed unpaired Student's t-test was used if only two conditions or genotypes were compared. In all other cases, one- or two-way ANOVAs followed by Tukey's, Dunnett's, or Sidak's multiple comparisons tests were employed, as indicated in the figure legends. p<0.05 was considered to be statistically significant. No samples or data were omitted during the analyses.

## Data availability

RNA-sequencing data have been deposited in the ENA database under accession number PRJEB20661.

## Acknowledgements

We thank all members of the Suter lab, especially Dr. Deniz Gökbuget, for discussions, Drs. Monica Ghidinelli and Ned Mantei for critically reading the manuscript, and Joanne Jeker, Francesco Santarella, the Functional Genomic Center Zürich (FGCZ), and the ScopeM imaging facility of ETH Zürich for excellent technical support. We also thank Dr. Dies Meijer (University of Edinburgh) for $Dhh^{Cre}$ mice and antibodies, Drs. Laura Feltri and Lawrence Wrabetz (University of Buffalo) for $Mpz^{Cre}$ mice, and Drs. Michael Hall and Markus Rüegg (University of Basel) for floxed-$Rptor$ mice.

## Additional information

### Funding

| Funder | Grant reference number | Author |
| --- | --- | --- |
| Schweizerischer Nationalfonds zur Förderung der Wissenschaftlichen Forschung | | Ueli Suter |
| European Commission | Marie Curie Actions Intra-European Fellowship | Camilla Norrmén |

The funders had no role in study design, data collection and interpretation, or the decision to submit the work for publication.

### Author contributions

Gianluca Figlia, Conceptualization, Data curation, Formal analysis, Investigation, Methodology, Writing—original draft, Writing—review and editing; Camilla Norrmén, Conceptualization, Formal analysis, Funding acquisition, Investigation, Methodology, Writing—review and editing; Jorge A Pereira, Conceptualization, Data curation, Formal analysis, Investigation, Methodology, Writing—review and editing; Daniel Gerber, Conceptualization, Formal analysis, Investigation, Methodology, Writing—review and editing, Designed graphics of Figure 7; Ueli Suter, Conceptualization, Resources, Data curation, Formal analysis, Supervision, Funding acquisition, Project administration, Writing—review and editing

Author ORCIDs
Gianluca Figlia ⓘ https://orcid.org/0000-0001-8689-8488
Ueli Suter ⓘ http://orcid.org/0000-0002-9211-5184

Ethics
Animal experimentation: Mice were kept at 12 hr light/dark cycle. All experiments were approved by the Veterinary Office of the Canton of Zurich, Switzerland (reference No. ZH161/2014, ZH44/2013)

Decision letter and Author response
Decision letter https://doi.org/10.7554/eLife.29241.035
Author response https://doi.org/10.7554/eLife.29241.036

---

# Additional files

## Supplementary files

• Supplementary file 1. Full-length western blot images overlaid with the corresponding membranes. (a) Western blot in *Figure 1a*. The membrane was cut as indicated by the continuous line and probed with the indicated antibodies. (b) Western blot in *Figure 1k*. The membrane was cut as indicated by the continuous line and probed with the indicated antibodies.
DOI: https://doi.org/10.7554/eLife.29241.023

• Supplementary file 2. Full-length western blot images overlaid with the corresponding membranes. (a) Western blot in *Figure 1—figure supplement 2c*. (b) Western blot in *Figure 2a*. The membrane was cut as indicated by the continuous line and probed with the indicated antibodies.
DOI: https://doi.org/10.7554/eLife.29241.024

• Supplementary file 3. Full-length western blot images overlaid with the corresponding membranes. Western blot in *Figure 2c*. The membrane was cut as indicated by the continuous line and probed with the indicated antibodies.
DOI: https://doi.org/10.7554/eLife.29241.025

• Supplementary file 4. Full-length western blot images overlaid with the corresponding membranes. (a) Western blot in *Figure 2—figure supplement 2a*. (b) Western blot in *Figure 2—figure supplement 2c*. The membrane was cut as indicated by the continuous line and probed with the indicated antibodies. (c) Western blot in *Figure 2—figure supplement 2b*. (d) Western blot in *Figure 2g*.
DOI: https://doi.org/10.7554/eLife.29241.026

• Supplementary file 5. Full-length western blot images overlaid with the corresponding membranes. (a) Western blot in *Figure 2k*. (b) Western blot in *Figure 4a*. The membrane was cut as indicated by the continuous line and probed with the indicated antibodies. Asterisks indicate unspecific bands. (c) Western blot in *Figure 3e*. (d) Western blot in *Figure 3f*.
DOI: https://doi.org/10.7554/eLife.29241.027

• Supplementary file 6. Full-length western blot images overlaid with the corresponding membranes. (a) Western blot in *Figure 5a*. The membrane was cut as indicated by the continuous line and probed with the indicated antibodies. (b) Western blot in *Figure 5b*. The membrane was cut as indicated by the continuous line and probed with the indicated antibodies.
DOI: https://doi.org/10.7554/eLife.29241.028

• Supplementary file 7. Full-length western blot images overlaid with the corresponding membranes. (a) Western blot in *Figure 5c*. The membrane was cut as indicated by the continuous line and probed with the indicated antibodies. (b) Western blot in *Figure 5d*. The membrane was cut as indicated by the continuous line and probed with the indicated antibodies.
DOI: https://doi.org/10.7554/eLife.29241.029

• Supplementary file 8. Full-length western blot images overlaid with the corresponding membranes. (a) Western blot in *Figure 6h*. (b) Western blot in *Figure 6j*. The membrane was cut as indicated by the continuous line and probed with the indicated antibodies.
DOI: https://doi.org/10.7554/eLife.29241.030

• Transparent reporting form

DOI: https://doi.org/10.7554/eLife.29241.031

### Major datasets

The following dataset was generated:

| Author(s) | Year | Dataset title | Dataset URL | Database, license, and accessibility information |
|---|---|---|---|---|
| Gianluca Figlia, Ueli Suter | 2017 | Dual Function of the PI3K-Akt-mTORC1 Axis in Myelination of the Peripheral Nervous System | http://www.ebi.ac.uk/ena/data/view/PRJEB20661 | Publicly available at the EBI European Nucleotide Archive (accession no: PRJEB20661) |

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
