## [Decision Letter]

Thank you for submitting your article "Dual Function of the PI3K-Akt-mTORC1 Axis in Myelination of the Peripheral Nervous System" for consideration by *eLife*. Your article has been reviewed by two peer reviewers, and the evaluation has been overseen by a Reviewing Editor and Gary Westbrook as the Senior Editor. The following individuals involved in review of your submission have agreed to reveal their identity: Robert H Miller (Reviewer #1); Wendy B. Macklin (Reviewer #2). The reviewers have discussed the reviews with one another and the Reviewing Editor has drafted this decision to help you prepare a revised submission.

Summary of reviews and reviewer/editor discussion:

The reviewers find that this is a compelling paper that investigates the functional roles of mTOR1 activation in Schwann cell myelination during different stages of development, homeostasis, and in the context of remyelination after injury. Using multiple genetic manipulations to disrupt key inhibitory components of mTORC1 (TSC1 or PTEN), the authors convincingly demonstrate, that hyperactivation of mTORC during early development blocks myelination through inhibition of Schwann cell differentiation and generation of radial sorting. By contrast, hyperactivation of mTORC in the adult animal after peripheral nerve crush results in hypermyelination. The studies also provide evidence that high mTORC1 activity inhibits Krox20 expression, a known master regulator of Schwann cell myelination, and demonstrate Krox20 as an effector of mTORC signaling in PNS myelination that is mediated by SK6.

Strengths of the study are the careful use of multiple genetic approaches to delineate the mechanisms by which mTORC modulates Schwann cell development and myelination in different contexts. The study also helps explain conflicting results raised by the series of earlier studies and defines a model that is important for future investigations on Schwann cell differentiation and remyelination. The results are well communicated, and data is of high quality and supports the interpretations. However there were a few points that require clarification and expanded discussion, as outlined below. (The full reviews are also appended below).

Essential revisions:

1) P0-PTEN-KO mice have fewer myelinated axons at P5, and Dhh-PTEN cKO mice are comparable, indicating that at either stage of differentiation, PTEN cKO/Akt overactivation induces decreased numbers of myelinated axons. However CNP-PTEN cKO mice have significantly more myelinated axons at 3 M (Goebbels et al. J Neurosci 2010), and that promoter should have deleted PTEN in this same time frame of Schwann cell differentiation. Some discussion of this is needed.

2) In the nerve injury experiments, according to their own data, the mice would have had increased myelin, altered Remak bundles and increased numbers of myelinated axons before the nerve injury. How does this impact the eventual myelin loss/recovery after nerve injury? How do they define "intact appearing myelin profiles" (Figure 6) in the environment of altered myelin?

*Reviewer #1:*

This is an excellent manuscript that clearly defines a complex phenomena in the mTORC modulation of PNS development. The authors have been careful to rule out other potential signaling paths and have, in many cases, provided multiple studies to support their interpretation. The manuscript represents one of the most detailed and careful studies of the role of mTORC signaling in peripheral myelination to date. There are few concerns with the data which is of high quality and supports the interpretation of the authors.

Overall, this is a very comprehensive and extremely well done manuscript that will be of interest to a wide range of developmental neurobiologists.

*Reviewer #2:*

This study by Figlia et al. focuses on an important research problem that has had significant, apparently contradictory data for a number of years. It has been clearly shown that mTORC1 activity is required for myelination. Yet, Dr. Suter's group and others have shown that deleting TSC1 from myelinating cells, which clearly upregulates mTORC1 activity since TSC1 normally inhibits mTORC1 activity, actually results in hypomyelination. The current manuscript is a detailed, complex analysis of why these unexpected data have been obtained by multiple investigators. This study is important and thorough.

This study demonstrates yet again that deleting TSC in Dhh-expressing Schwann cells, i.e., in early Schwann cell lineage cells, results in increased mTORC1 activity yet hypomyelination, and that rapamycin treatment induces partial recovery of myelination. They then demonstrate that deletion of PTEN, which increases mTORC1 activity by way of activated Akt signaling, in either Dhh or P0-expressing cells results in a partial reduction in myelination. The double TSC/PTEN conditional knockout eliminates essentially all myelination. Rapamycin can recover some of this loss of myelination. Thus, these data are convincing and consistent with some of the data in the literature that increasing mTOR signaling by deletion of either endogenous inhibitor can reduce myelination in Schwann cells.

The studies also show convincingly that high mTORC1 activity likely inhibits Krox20 expression, consistent with its apparent inhibition of initial Schwann cell differentiation into myelinating Schwann cells.

They demonstrate, as has Goebbels et al., 2010, that deleting PTEN and other mTORC1 activators in adults induces hypermyelination. This is a far more in-depth analysis of sciatic nerve hypermyelination than in the Goebbels et al. study, and they go on to study remyelination after injury in these mTORC1-activated mice. Importantly, they show that, as in development, remyelination is impaired in mice with excess mTORC1 signaling in remyelinating Schwann cells.

This study is important as it answers many of the questions raised by the series of earlier studies, and defines a model that is important for future investigations on Schwann cell differentiation.

Nevertheless, there are some important concerns:

1) P0-PTEN-KO mice have fewer myelinated axons at P5, and Dhh-PTEN cKO mice are comparable, indicating that at either stage of differentiation, PTEN cKO/Akt overactivation induces decreased numbers of myelinated axons. However CNP-PTEN cKO mice have significantly more myelinated axons at 3 M (Goebbels et al. J Neurosci 2010), and that promoter should have deleted PTEN in this same time frame of Schwann cell differentiation. Some discussion of this is needed.

2) In the nerve injury experiments, according to their own data, the mice would have had increased myelin, altered Remak bundles and increased numbers of myelinated axons before the nerve injury. How does this impact the eventual myelin loss/recovery after nerve injury? How do they define "intact appearing myelin profiles" (Figure 6) in the environment of altered myelin?

---

## [Author Response]

Essential revisions:

1) P0-PTEN-KO mice have fewer myelinated axons at P5, and Dhh-PTEN cKO mice are comparable, indicating that at either stage of differentiation, PTEN cKO/Akt overactivation induces decreased numbers of myelinated axons. However CNP-PTEN cKO mice have significantly more myelinated axons at 3 M (Goebbels et al. J Neurosci 2010), and that promoter should have deleted PTEN in this same time frame of Schwann cell differentiation. Some discussion of this is needed.

In the study cited by the reviewers, conditional ablation of PTEN in the Schwann cell lineage increased the number of myelinated fibers in 3 months-old mice, most likely due to aberrant myelination of normally non-myelinated C fibers. However, our results show that physiologically high activity of the PI3K-Akt-mTORC1 axis before the ordinary developmental onset of myelination has a myelination-inhibitory function. Thus, chronic hyperactivation of PI3K-Akt signaling may be able to, in the long term, drive Remak cells to a myelinating-like phenotype. We discuss this possibility now in the fourth paragraph of the Discussion.

*2) In the nerve injury experiments, according to their own data, the mice would have had increased myelin, altered Remak bundles and increased numbers of myelinated axons before the nerve injury. How does this impact the eventual myelin loss/recovery after nerve injury? How do they define "intact appearing myelin profiles" (Figure 6) in the environment of altered myelin?*

We agree with the reviewers that the observed radial hypermyelination in inducible TSC1 and PTEN mutants could have an impact on the events following nerve injury. This is a limitation of the remyelination studies in our work. However, the lack of major differences between mutants and controls in the immediate demyelination phase after nerve crush indicates that the defective remyelination of TSC1 and PTEN mutants at later time points is probably not due to the previous “maintenance phenotype”. We thank the reviewers for pointing this limitation out to us. We have ameliorated the manuscript accordingly (subsection “The differentiation status of SCs determines the outcome of mTORC1 activation”, second paragraph). We also added a definition of intact-appearing myelin profiles as “non-discontinuous and non-collapsed myelin rings”.